# Novel mathematical approach to accurately quantify 3D endothelial cell morphology and vessel geometry based on fluorescently marked endothelial cell contours: Application to the dorsal aorta of wild-type and Endoglin-deficient zebrafish embryos

**Daniel Seeler**[1,2,3], **Nastasja Grdseloff**[2¤], **Claudia Jasmin Rödel**[2], **Charlotte Kloft**[4], **Salim Abdelilah-Seyfried**[2], **Wilhelm Huisinga**[1,2]*

**1** Faculty of Science, Institute of Mathematics, University of Potsdam, Potsdam, Germany, **2** Faculty of Science, Institute of Biochemistry and Biology, University of Potsdam, Potsdam, Germany, **3** PharMetrX Graduate Research Training Program: Pharmacometrics & Computational Disease Modelling, Berlin/Potsdam, Germany, **4** Department of Biology, Chemistry, and Pharmacy, Institute of Pharmacy, Freie Universität Berlin, Berlin, Germany

¤ Current address: HeartBeat.bio AG, Vienna BioCenter, Vienna, Austria
* huisinga@uni-potsdam.de

## Abstract

Endothelial cells, which line the lumen of blood vessels, locally sense and respond to blood flow. In response to altered blood flow dynamics during early embryonic development, these cells undergo shape changes that directly affect vessel geometry: In the dorsal aorta of zebrafish embryos, elongation of endothelial cells in the direction of flow between 48 and 72 hours post fertilization (hpf) reduces the vessel's diameter. This remodeling process requires Endoglin; excessive endothelial cell growth in the protein's absence results in vessel diameter increases. To understand how these changes in vessel geometry emerge from morphological changes of individual endothelial cells, we developed a novel mathematical approach that allows 3D reconstruction and quantification of both dorsal aorta geometry and endothelial cell surface morphology. Based on fluorescently marked endothelial cell contours, we inferred cross-sections of the dorsal aorta that accounted for dorsal flattening of the vessel. By projection of endothelial cell contours onto the estimated cross-sections and subsequent triangulation, we finally reconstructed 3D surfaces of the individual cells. By simultaneously reconstructing vessel cross-sections and cell surfaces, we found in an exploratory analysis that morphology varied between endothelial cells located in different sectors of the dorsal aorta in both wild-type and Endoglin-deficient zebrafish embryos: In wild-types, ventral endothelial cells were smaller and more elongated in flow direction than dorsal endothelial cells at both 48 hpf and 72 hpf. Although dorsal and ventral endothelial cells in Endoglin-deficient embryos had similar sizes at 48 hpf, dorsal endothelial cells were much larger at 72 hpf. In Endoglin-deficient embryos, elongation in flow direction increased between 48 hpf and 72 hpf in ventral endothelial cells but hardly changed in dorsal

**Data Availability Statement:** The analysis' underlying microscopy images including the manually annotated EC contours are available from the Zenodo database via https://doi.org/10.5281/zenodo.11353016. The code, all intermediate results, and the values and statistics underlying the shown figures are available from the Zenodo database via https://doi.org/10.5281/zenodo.10549101.

**Funding:** D.S. kindly acknowledges financial support from the Graduate Research Training Program PharMetrX: Pharmacometrics & Computational Disease Modelling, Berlin/Potsdam, Germany. S.A.-S. was generously supported by SFB958, Deutsche Forschungsgemeinschaft (DFG) projects SE2016/7-3, SE2016/10-1, SE2016/13-1, the Leducq Transatlantic Network of Excellence "21CVD03 - ReVAMP", and the Marie Skłodowska-Curie Innovative Training Network (ITN) "V.A. Cure". The funders had no role in study design, data collection and analysis, decision to publish, or preparation of the manuscript.

**Competing interests:** The authors have declared that no competing interests exist.

endothelial cells. Hereby, we provide evidence that dorsal endothelial cells contribute most to the disparate changes in dorsal aorta diameter in wild-type and Endoglin-deficient embryos between 48 hpf and 72 hpf.

## Author summary

Endothelial cells, which form the innermost layer of each blood vessel, sense and respond to blood flow. During early embryonic development in zebrafish, endothelial cells of the dorsal aorta elongate in the direction of blood flow and hereby decrease the vessel's diameter. To understand how these changes in vessel geometry emerge from morphological changes of individual endothelial cells, it is critical to precisely quantify both vessel geometry and cell morphology. To this end, we developed a novel mathematical approach that leverages information from fluorescently marked endothelial cell contours to reconstruct the dorsal aorta's surface in 3D. In an exploratory analysis, we applied this method to wild-type and mutant zebrafish embryos lacking functional Endoglin that is required for the physiological vessel diameter decrease. By quantifying vessel geometry and cell morphology in these embryos, we found that size and elongation in the direction of blood flow varied between endothelial cells located in different vessel sectors. Notably, we determined a subgroup of endothelial cells that contributed most to the vessel diameter increases in the absence of Endoglin. Future studies can investigate whether variability in endothelial cell behavior also contributes to the onset of human vascular malformations occurring due to a loss of Endoglin.

## Introduction

During embryonic development and adult life, the vasculature is continuously reshaped to maintain sufficient nutrient supply to tissues and to regulate systemic blood pressure. Endothelial cells (ECs), which line the lumen of blood vessels, can locally sense [1, 2] long-term changes of blood flow dynamics, occurring over a time span ranging between hours during embryogenesis [3] and weeks to months during adult life [4, 5]. As a consequence, vascular cells, i.e., ECs and mural cells, undergo behavioral changes that include alterations of cell shape, cellular rearrangements, migration, proliferation and cell death [6–8]. The combination of these cellular processes and structural alterations of the extracellular matrix results in the generation of new vessels via angiogenesis, the regression of weakly perfused vessels and changes in vessel luminal diameter; these coupled processes are called vascular remodeling [6–9]. It is crucial to improve our understanding of EC behavior during vascular remodeling as aberrations in sensing or responding to biochemical stimulation by blood flow on the level of ECs can cause diseases like atherosclerosis or a number of vascular anomalies [10–12].

Vascular remodeling can be precisely studied in zebrafish embryos, as their transparency and external development allows for non-invasive live-imaging of their vasculature over time. Additionally, many transgenic reporters are available that enable EC-specific fluorescent labeling of various cellular compartments [13]. The dorsal aorta (DA) is the major artery in the zebrafish trunk and undergoes drastic remodeling during embryonic development (for an illustration, see S7 Fig in [3]): Between 48 hours post fertilization (hpf) and 72 hpf, ECs in the DA elongate in the direction of blood flow. As a consequence, the vessel length increases, but its diameter decreases. It was found that the TGF-beta co-receptor Endoglin is necessary for

this remodeling process: Although ECs in Endoglin-deficient embryos also elongate in the direction of flow between 48 hpf and 72 hpf, excessive EC growth in the protein's absence causes an increase in DA diameter instead [3].

To better understand the interplay of vessel geometry and cell morphology during vascular remodeling of the DA, it is critical to precisely quantify both DA geometry and EC surface morphology in the same embryo. While quantities like vessel diameter and EC perimeter can be directly measured in 3D microscopy images, mathematical approaches are required for the computation of other morphometric measures like the surface area of ECs, which surround the blood vessel. To learn how the disparate changes in DA diameter in wild-type and Endo-glin-deficient zebrafish embryos emerge from EC shape changes, Sugden et al. computed 2D representations of ECs from their fluorescently labeled 3D cell contours [3]. For each EC *individually*, they first estimated an elliptic or hyperbolic cylinder from its contour. They then projected the EC contour onto the estimated cylinder and unrolled it. Hereby, the authors were able to quantify morphology of the unrolled ECs in 2D. This method, however, is not designed to compute an accurate description of vessel geometry: Cylinders estimated from different ECs whose surfaces contribute to the same vessel cross-section can differ substantially. Furthermore, by computation of a single cylinder per EC contour, variation in cross-sectional shape along the length of the cell is neglected.

To analyze the environment of ECs undergoing endothelial-to-hematopoietic transition (EHT) in the DA, Lancino et al. computed a 2D representation of the DA from fluorescently labeled cell contours [14]. Along the length of a DA segment, they estimated circular cross-sections with variable centers but fixed radius that best fit the local fluorescent cell contour signal. To further account for variation in cross-sectional shape along the vessel axis, points on estimated circles were assigned the locally maximal fluorescence intensities in radial direction. The maximal fluorescence intensities were then unrolled onto a rectangle whose width was given by the DA's length and its height by the cross-sectional circumference. While the unrolled map enabled the authors to visualize and quantify cell connectivity, it is not suited for precise analysis of either vessel geometry or cell surface morphology: By projecting fluorescence intensities onto a circle with fixed radius, variations in cross-sectional geometry in radial direction are dismissed. Furthermore, by changing the centers of the DA's cross-sections along the vessel axis without incorporating these translations in the planar representation, cell contours become distorted. Quantification of morphology from complex cell contours can thus result in drastic deviations, e.g., in cell perimeter.

Finally, to study the spatial distribution of ECs within DA cross-sections, Campinho et al. reconstructed the 3D apico-luminal surface of the DA from fluorescent endothelial cytoskeleton markers [15, 16]. Along the DA's length, they estimated elliptic cross-sections from local endothelial cytoskeletal signals. They then projected the centers of fluorescently labeled EC nuclei onto the reconstructed tube. While this method allowed the authors to quantify both the luminal area of the DA along its axis and the distribution of EC nuclei in different cross-sectional sectors of the DA, the authors did not leverage information from EC contours. Without this information, the boundaries of EC surfaces on the estimated vessel surface cannot be defined and thus, the individual cells' surface morphology cannot be evaluated. While all the aforementioned approaches described DA cross-sections using circles or ellipses [3, 14, 15], we repeatedly observed in 3D microscopy images that dorsal segments of DA cross-sections can be flattened in comparison to their ventral segments (see S1 Fig).

To simultaneously and physiologically accurately describe cross-sectional geometry and EC surface morphology in the DA, we developed a novel mathematical approach that enables reconstruction of the apico-luminal surface of the vessel's EC layer. By estimating both vessel cross-sections and EC surfaces solely from EC contours, our approach guarantees consistency

of vessel geometry and EC surface morphology. To account for the observed dorsal flattening of the DA, we developed a novel parametric cross-sectional shape model with inherent dorsal-ventral asymmetry. Following vessel surface reconstruction, cross-sectional geometry of the DA and EC surface morphology can be quantified within the same embryo. By quantification of EC surface morphology in 3D, we avoid cartographic distortions that would be caused by unrolling of the changing cross-sections along the vessel axis. Our preprocessing and surface reconstruction methods require little user intervention and thus enable statistical analyses of cross-sectional geometry and EC surface morphology over large numbers of embryos.

Using our mathematical approach, we quantified DA geometry and EC surface morphology in wild-type and Endoglin-deficient zebrafish embryos at 48 hpf and 72 hpf and found good agreement with the results reported in [3]. Due to the precise reconstruction of cross-sectional geometry and EC surface morphology in the same blood vessels, our method further enabled us to identify a previously unrecognized dorsal-ventral asymmetry of EC surface morphology in the dorsal aorta: In wild-type embryos, ventral ECs were smaller and more elongated in the direction of blood flow than dorsal ECs at both 48 hpf and 72 hpf. Although dorsal and ventral ECs in Endoglin-deficient embryos had similar sizes at 48 hpf, dorsal ECs were much larger at 72 hpf. In Endoglin-deficient embryos, elongation in the direction of flow increased between 48 hpf and 72 hpf in ventral ECs but hardly changed in dorsal ECs. Hereby, we provide evidence that dorsal ECs contribute most to the disparate changes in dorsal aorta diameter in wild-type and Endoglin-deficient embryos between 48 hpf and 72 hpf.

## Materials and methods

### Ethics statement

Handling of zebrafish was done in compliance with German and Brandenburg state law. The permission for breeding and keeping of vertebrates according to Section 11 (1) sentence 1 no. 1 of the law for the protection of animals (Tierschutzgesetz) was obtained on 24 February 2015 and is carefully monitored by the local authority for animal protection (LAVG, Brandenburg, Germany) and in full accordance with the European Directive 2010/63/UE regarding the protection of animals used for scientific purposes.

### Data acquisition

We reconstructed DA geometry and EC surface morphology from manually annotated EC contours in 7 wild-type (wt) and 6 Endoglin-deficient (Eng-def) zebrafish embryos at 48 hpf and 72 hpf (see Table 1, analysis data). For validation of our mathematical approach, we acquired independent data consisting of two wild-type embryos at 72 hpf where the vessel lumen was visualized by angiography (see Table 1, validation data). To allow quantification of annotation uncertainty, we annotated each EC contour twice in both of these embryos.

**Table 1. Data used for the analysis of dorsal aorta geometry and endothelial cell morphology and for validation.**

| Data | Purpose | Phenotype | Time point | Embryo IDs | EC contours | Angiography |
|---|---|---|---|---|---|---|
| analysis data | geometric and morphometric analysis | wt[†] | 48 hpf, 72 hpf | 1–7 | annotated once | no |
| | | Eng-def[‡] | | 8–13 | | |
| validation data | validation of methodology | wt | 72 hpf | 14–15 | annotated twice | yes |

[†]: Wild-type (wt) embryos carry maximally one $eng^{mu130}$ loss-of-function allele.

[‡]: Endoglin-deficient (Eng-def) embryos carry two $eng^{mu130}$ loss-of-function alleles.

**Zebrafish genetics and maintenance.** The following strains of zebrafish were maintained under standard conditions as previously described [17]: Zebrafish with the transgenic reporter *Tg(fli1:pecam1-EGFP)*[ncv27] express the fusion protein Pecam1-EGFP specifically in ECs due to the *fli1* promoter; fusion of Pecam1 to EGFP fluorescently marks endothelial cell-cell junctions [18]. In zebrafish with the transgenic reporter *Tg(fli1:NLS-mCherry)*[ubs10], nuclear localization of the fusion protein NLS-mCherry fluorescently marks EC nuclei [19]. Finally, *eng*[mu130] zebrafish contain a frameshift mutation in the *eng* gene that results in a premature stop codon; nonsense-mediated decay of *eng*[mu130] mRNA leads to a loss of function of the encoded Endoglin protein [3].

To allow identification of EC contours, all the zebrafish analyzed in this article expressed at least one allele of *Tg(fli1:pecam1-EGFP)*[ncv27]. Co-expression of one or two alleles of *Tg(fli1: NLS-mCherry)*[ubs10] in a subset of the analyzed zebrafish further aided in the visual separation of neighboring ECs. Due to the recessive phenotype of the *eng*[mu130] allele [3], we refer to embryos homozygous for the loss-of-function mutation *eng*[mu130] as Endoglin-deficient embryos and to all other embryos as wild-types.

**Microangiography and live imaging.** For live imaging experiments, embryos were collected, incubated at 28.5°C and treated at 24 hpf with 1-phenyl-2-thiourea (PTU) (Sigma Aldrich) to inhibit pigmentation. Embryos were manually dechorionated, anesthetized by incubation in egg water containing 0.003% Tricaine and embedded laterally in 1% low melting agarose (Lonza 50081) containing 0.16 mg/mL Tricaine in glass-bottom dishes (MatTek) and imaged. For microangiography, anesthetized embryos were injected with 5 μg/μL Dextran Texas Red (70 kDa, D1864, Invitrogen) into the common cardinal vein at 48 hpf and immediately processed for imaging. Subsequently, embryos were recovered, incubated at 28.5°C in normal egg water, remounted at 72 hpf and imaged again. After imaging, all embryos were genotyped by PCR.

Imaging was conducted with an LSM 780 confocal microscope (Zeiss) using a 20x-objective. As dechorionation flattened the embryos' overall morphology, they naturally lay on either side in the microscope. Imaging them from the lateral side also ensured least tissue interference for confocal microscopy. Due to the bilateral symmetry of the DA and the surrounding tissue, information gained by imaging from either the left or right side is equivalent. In this study, we thus chose to place all embryos on their right side in the microscope.

## General notation

Throughout the article, we denote the ordered sequence of 3D points of endothelial cell contour $i = 1, 2, \ldots, N$ that is the result of computational step $\ell$ by $\mathrm{EC}_{\ell,i}$. For example, $\mathrm{EC}_{\mathrm{anno},i}$ is the manually annotated contour of endothelial cell $i$. Each sequence $\mathrm{EC}_{\ell,i}$ consists of $n_{\ell,i}$ points:

$$\mathrm{EC}_{\ell,i} = (p_1, p_2, \ldots, p_{n_{\ell,i}}), \tag{1}$$

where we write the $x$, $y$ and $z$ components of $p \in \mathrm{EC}_{\ell,i}$ as

$$p = \left([p]_x, [p]_y, [p]_z\right)^\top \in \mathbb{R}^3, \tag{2}$$

$$[p]_{xy} = \left([p]_x, [p]_y\right)^\top \in \mathbb{R}^2. \tag{3}$$

We denote the successor/predecessor of a point $p \in EC_{\ell,i}$ using $p^+$ and $p^-$, respectively:

$$p_1^+ = p_2; \qquad p_2^+ = p_3; \qquad \ldots; \qquad p_{n_{\ell,i}}^+ = p_1, \tag{4}$$

$$p_{n_{\ell,i}}^- = p_{n_{\ell,i}-1}; \qquad p_{n_{\ell,i}-1}^- = p_{n_{\ell,i}-2}; \qquad \ldots; \qquad p_1^- = p_{n_{\ell,i}}. \tag{5}$$

Concatenating the points of all cell contours ($i = 1, 2, \ldots, N$), an ordered sequence

$$EC_\ell = EC_{\ell,1} | EC_{\ell,2} | \ldots | EC_{\ell,N} \tag{6}$$

of length $\sum_{i=1}^{N} n_{\ell,i}$ is obtained. Here, | denotes concatenation of tuples. Throughout the text, we use the multiplication sign both for the multiplication $\alpha \cdot v$ of a scalar $\alpha \in \mathbb{R}$ and a vector $v \in \mathbb{R}^3$ and for the inner product $v_1 \cdot v_2 = [v_1]_x \cdot [v_2]_x + [v_1]_y \cdot [v_2]_y + [v_1]_z \cdot [v_2]_z$ of two vectors $v_1, v_2 \in \mathbb{R}^3$. The type of multiplication is clear from the dimensions of the factors.

We now explain the method for reconstruction and quantification of DA geometry and EC surface morphology from fluorescently marked ECs. An overview of the method is shown in Fig 1. Following description of each computational step with the introduction of all involved tuning parameters, we explain how we evaluated the goodness of fit of our approach, how we chose suitable tuning parameter values and, finally, how we validated our novel mathematical approach. To increase accessibility of the article, we provide a list of frequently used mathematical symbols in S1 Table.

## Step 1: Preprocessing of fluorescent images

**a) Manual annotation of endothelial cell contours.** To obtain EC contours within the DA, we employed Imaris Software V9.5.1 (Bitplane). Within a 3D reconstruction of the fluorescent signal from confocal z-stacks, we manually outlined Pecam1-EGFP-marked cell junctions using the "Measurements" tool (see S7(A) Fig). To reduce within-embryo and between-embryo variability, we annotated cell contours always within the same vessel segment in each embryo; the annotated segment is positioned posterior to the tip of the yolk extension and ranges over 3–4 intersegmental vessels. During outlining, we excluded cells if they were located at the ventral most position of the DA and their contours, i.e., their cell junctions, were perfectly round and only circumscribed a small area. As these cells are likely (precursor) hematopoietic stem and progenitor stem cells (HSPCs) and would shortly extrude from the DA [14], they do not contribute to vessel geometry over a substantial time span. We confirmed that points were placed in the center of the fluorescent signal by 3D image rotations. Each EC was annotated independently of its neighbors, i.e., joint cell-cell contacts were annotated twice. This increased the number of points available per cell-cell contact. We ensured that contour segments between consecutive points were well approximated by a straight line. This reduced the number of points needed to accurately describe EC contours. Additionally, we annotated vectors, defined as two ordered points, that provided approximate directions of anterior-posterior, dorsal-ventral and left-right axes (see S7(B) Fig). Data were exported as CSV files containing the points' 3D coordinates. We denote the manually annotated contour of endothelial cell $i$ as $EC_{\text{anno},i}$.

For validation of our mathematical approach against angiography, we additionally outlined cross-sections of the DA within 3D reconstructions of the fluorescent signal from confocal z-stacks of Dextran-perfused DAs. Using the "Ortho slicer" tool, we manually annotated the outlines of several cross-sections along the anterior-posterior axis of the DA with the "Measurements" tool (see S7(C) Fig). We confirmed that points of a cross-section lay approximately on a plane by 3D rotation. Again, we ensured that contour segments between consecutive points

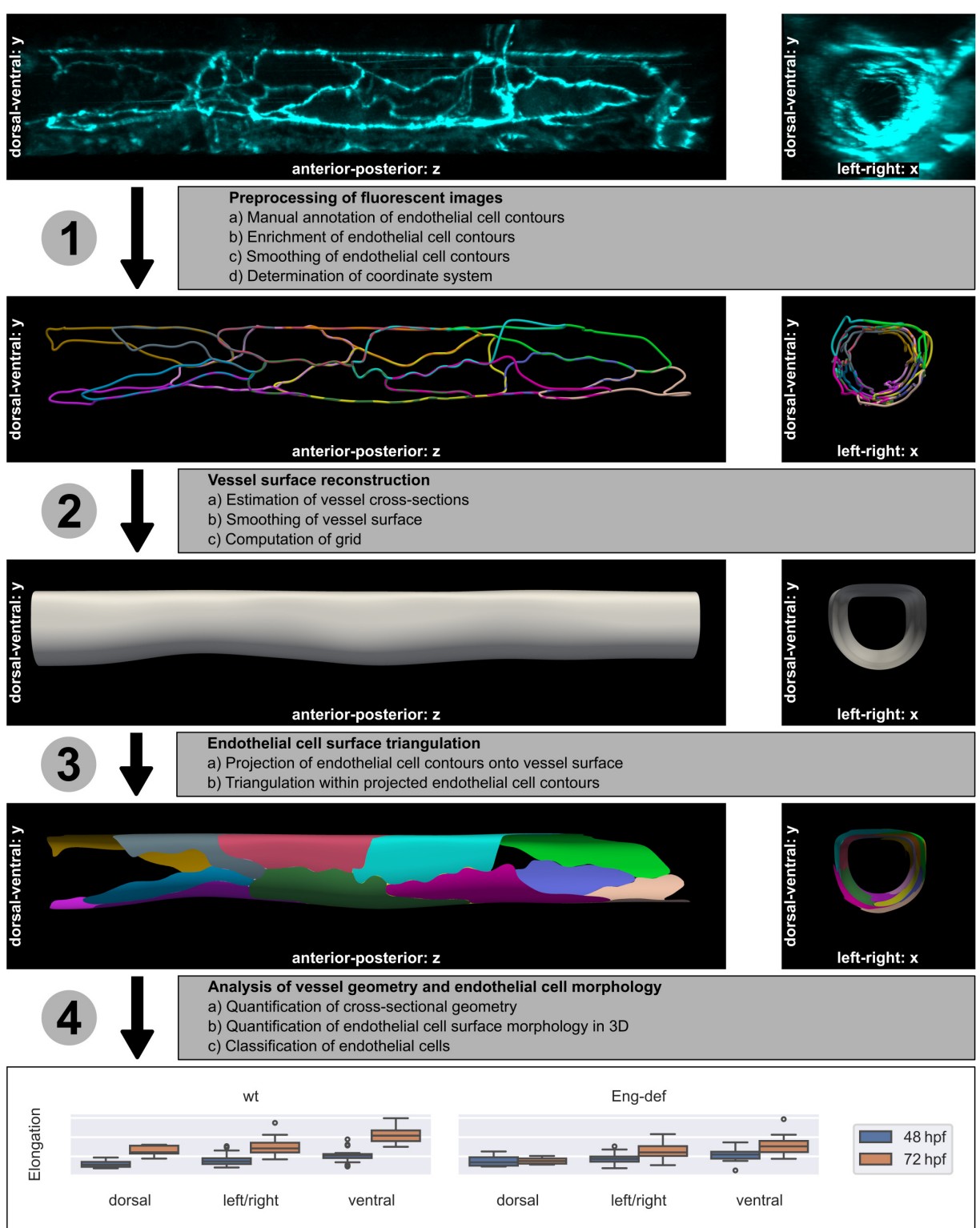

**Fig 1. Method overview.** The individual steps are explained in the text. The 3D microscopy image in the top row stems from Endoglin-deficient embryo 13 at 72 hpf and was used for the subsequent computations in the rows below. Elongation of ECs was quantified using the analysis data (see Table 1).

**Fig 2. Cell contour enrichment.** (A) Enrichment by neighboring cells. $EC_{enri,i}$ is the current version of the enriched cell contour of cell $i$. $EC_{anno,i'}$ is the manually annotated cell contour of cell $i'$. In the previous iteration, it was checked whether point $q^- \in EC_{anno,i'}$ lies in any cylinder centered around an edge of $EC_{enri,i}$. As $q^-$ was too far away from these edges, it was not inserted into $EC_{enri,i}$. In the current iteration, it is found that $q \in EC_{anno,i'}$ lies in the cylinder centered around the edge $(p, p^+)$ of $EC_{enri,i}$ with radius $r$ and length $h = ||p^+ - p||_2 + 2\Delta h$. Thus, $q$ is inserted into $EC_{enri,i}$. Following insertion of $q$, point $q^+$ will be inserted into $EC_{enri,i}$ in the subsequent iteration because it will be located within the cylinder centered around $(q, p^+)$. (B) Enrichment by interpolation. The enriched cell contour $EC_{enri,i}$ resulting from adding both $q$ and $q^+$ is shown. Two points $q_p(1)$ and $q_p(2)$ obtained by linear interpolation are added on the edge $(p, q)$.

were well approximated by a straight line. In the remainder of the text, we refer to the outlined DA cross-sections as angiogram slices.

**b) Enrichment of endothelial cell contours.** As we annotated each EC independently of its neighbors, joint cell-cell contacts were annotated twice. To leverage this information in the inference process, we enriched each cell contour by points of its neighbors in an iterative process (see Fig 2A): We initialized the enriched cell contour $EC_{enri,i}$ of cell $i$ with its manually annotated cell contour $EC_{anno,i}$. Then, we iterated over all points $q \in EC_{anno}$ with $q \notin EC_{anno,i}$. Whenever a point $q$ lay in at least one cylinder centered around an edge $(p, p^+)$ of the current version of the enriched contour $EC_{enri,i}$, we inserted $q$ into $EC_{enri,i}$. To control which points were inserted, we defined two tuning parameters: the cylinder radius $r > 0$ and the length extension $\Delta h > 0$ resulting in a cylinder length $h = ||p^+ - p||_2 + 2\Delta h$ for contour edge $(p, p^+)$.

Since we annotated fewer points in straight contour segments, the density of annotated points varied along contours. To homogenize this point density, we added $n_{interp}(p)$ additional points on each contour edge $(p, p^+)$ by linear interpolation (see Fig 2B). To ensure that the point density on all edges was comparable, we scaled the number $n_{interp}(p)$ of points interpolated on contour edge $(p, p^+)$ with the edge length:

$$n_{interp}(p) = \left\lfloor n_{interp,rel} \cdot ||p^+ - p||_2 \right\rfloor, \tag{7}$$

where $n_{interp,rel}$ is a tuning parameter controlling the relative number of points interpolated per unit length (here: μm) on edges of any contour. We provide further details on the cell contour enrichment in Section 1 of S1 Appendix. To see an exemplary outcome of cell contour enrichment, compare S8(A) and S8(B) Fig.

**c) Smoothing of endothelial cell contours.** To obtain a continuous representation of each cell contour, we employed cubic spline curves. Cubic splines are defined by piecewise polynomials of degree 3. These functions and their first and second derivatives are continuous and can thus accurately describe both concave and convex regions of cell contours. More specifically, we fitted periodic cubic smoothing splines [20]

$$s_i\colon [0, 1] \to \mathbb{R}^3; \quad i = 1, 2, \ldots, N \tag{8}$$

to the enriched cell contours. These curves minimize discontinuities in the third derivative while keeping the sum of squared distances between the points of the enriched contours and

their projections onto the splines below or equal to a user-specified threshold $\epsilon_{\text{spline},i} > 0$ (corresponding to $S$ in [20]). To ensure that splines fitted equally well to contours of small and large cells, we scaled the individual error thresholds $\epsilon_{\text{spline},i}$ by the number $n_{\text{enri},i}$ of points on the respective enriched cell contours:

$$\epsilon_{\text{spline},i} = \epsilon_{\text{spline}} \cdot n_{\text{enri},i}, \tag{9}$$

where $\epsilon_{\text{spline}}$ is a tuning parameter that serves as an upper bound on the mean squared distance for all splines. We parameterized each spline by the normalized arc length of the corresponding enriched cell contour.

Finally, we computed equidistant points on each endothelial cell contour spline. We denote the computed equidistant data points on the contour spline of cell $i$ as

$$\text{EC}_{\text{spline},i} = (p_1, p_2, ..., p_{n_{\text{spline},i}}), \tag{10}$$

where

$$0 = [p_1]_u < [p_2]_u < \ldots < [p_{n_{\text{spline},i}}]_u < 1 \tag{11}$$

are the parameter values of the points on the spline $s_i$. To homogenize the density of points on EC contours, i.e., to take into account that large cells contain more information on the vessel surface than small cells, we scaled the number $n_{\text{spline},i}$ of points computed on each cell contour spline with the cell's perimeter:

$$n_{\text{spline},i} = \left\lfloor n_{\text{spline,rel}} \cdot \int_0^1 ||s_i'(u)||_2 \, \mathrm{d}u \right\rfloor, \tag{12}$$

where $n_{\text{spline,rel}}$ is a tuning parameter that controls the relative number of points computed per unit length (here: μm) for all splines. For more details on the computation of equidistant points on cell contour splines, see Section 5.1 of S1 Appendix. To see an exemplary outcome of cell contour smoothing, compare S8(B) and S8(C) Fig.

**d) Determination of coordinate system.** The original coordinates of the cell contours depended on the position of the zebrafish embryo (and thus of the DA) within the mounting medium during imaging. As the DA is an approximately straight blood vessel without bifurcations, it is possible to define a straight line representing its anterior-posterior axis. Vessel cross-sections are then located on parallel planes perpendicular to this axis.

For each embryo at either time point, we defined the left-right ($x$), dorsal-ventral ($y$) and anterior-posterior ($z$) axes such that a single cross-sectional shape, termed mean shape, best fitted to all data points projected onto the $xy$-plane (see next section for the employed cross-sectional shape). The relationship between the coordinate system's axes and the mean shape are illustrated in S3 Fig. Our manually annotated vector for the dorsal-ventral axis was used to initialize the process of coordinate system determination. To ensure that the directionality of the anterior-posterior and left-right axis were not inverted, we compared the estimated axes with their respective manually annotated axis vectors. Below, the mean shape estimated together with the coordinate system is denoted $\bar{\varphi}$ with parameters $\bar{\theta}$. To differentiate the coordinates of contours in the original and the estimated coordinate system, we write $s_{\text{transf},i}$ for the spline of cell $i$ within the new coordinate system and $\text{EC}_{\text{transf},i}$ for the equidistant points on the spline of cell $i$ within the new coordinate system. Further details on the determination of the coordinate system and the mean shape are provided in Section 2 of S1 Appendix. The difference between the coordinate system given by the embryo's position within the mounting medium and the estimated coordinate system are exemplified in S9 Fig.

## Step 2: Vessel surface reconstruction

**a) Estimation of vessel cross-sections.** We estimated $M$ local cross-sectional shapes $\varphi_k(u)$ $:= \varphi(u; \theta_k)$ with parameters $\theta_k$, $k = 1, 2, \ldots, M$, at equidistant positions $z_1 < z_2 < \ldots < z_M$ on the anterior-posterior axis. To homogenize the density of estimated vessel cross-sections for different vessel segment lengths, we scaled the number $M$ of estimated cross-sectional shapes with the length of the segment on the anterior-posterior axis:

$$M = \left\lfloor M_{\text{rel}} \cdot \max_{p,q \in \text{EC}_{\text{transf}}} \left| [p]_z - [q]_z \right| \right\rfloor, \tag{13}$$

where $M_{\text{rel}}$ is a tuning parameter controlling the relative number of cross-sections estimated per unit length (here: μm) for all data sets, i.e., embryos at either time point.

Each cross-sectional shape $\varphi_k$ was initialized with the mean shape $\bar{\varphi}$. To account for the DA's dorsal-ventral asymmetry, we employed superelliptic cross-sectional shapes. The superellipse shape interpolates between an ellipse and a rectangle (see Fig 3A) and is thus very suitable to model dorsal flattening. By joining two (half) superellipses, we allowed that the ventral segment of the DA's cross-sections was not flattened. Our novel cross-sectional shape model

$$\varphi: [0, 2\pi] \rightarrow \mathbb{R}^2 \tag{14}$$

has parameters $\theta = (m_x, m_y, a, b, c, \alpha, \beta) \in \mathbb{R}^7$ (see Fig 3B and also Section 3.1 of S1 Appendix).

A difficulty of the estimation of cross-sectional shapes is that there exists no closed form solution of the euclidean distance of a given point to a general superellipse. To obtain a simple and computationally inexpensive approximation of the orthogonal projection $\Pi_\theta^{\text{shape}}([p]_{xy})$ of the $xy$-coordinates of cell contour point $p$ onto the cross-sectional shape $\varphi(u; \theta)$ with parameters $\theta$, we approximated the shape by a sufficiently dense polygon consisting of $n_{\text{poly}}$ points with linearly spaced parameter values (see also Section 3.2 of S1 Appendix).

We now explain the estimation of a local cross-sectional shape (see Fig 4 for an exemplary illustration). EC junction markers are only expressed on a small fraction of a blood vessel's apico-luminal surface. Locally, information on cell junctions hence provides insufficient information to estimate any type of cross-sectional shape. To be able to estimate vessel cross-

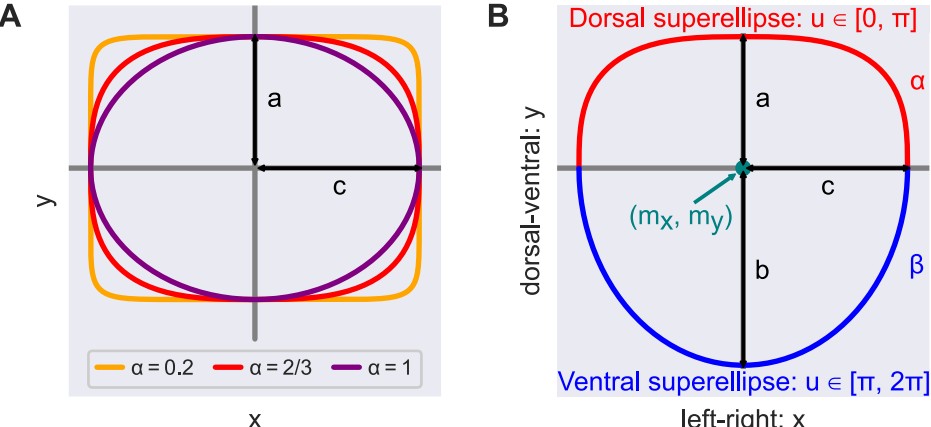

**Fig 3. Cross-sectional shape model. (A)** All points on the superellipses satisfy $|x/c|^{2/\alpha} + |y/a|^{2/\alpha} = 1$. For $\alpha = 1$ the superellipse reduces to an ellipse. The smaller the value of $\alpha < 1$, the closer the superellipse shape is to a rectangle. **(B)** We modeled DA cross-sections by joining two (half) superellipses. Here, $(m_x, m_y)$ is the shape's midpoint and $a, b, c$ are the shape's semi-axis lengths; $\alpha, \beta$ specify the extent of flattening of each superellipse.

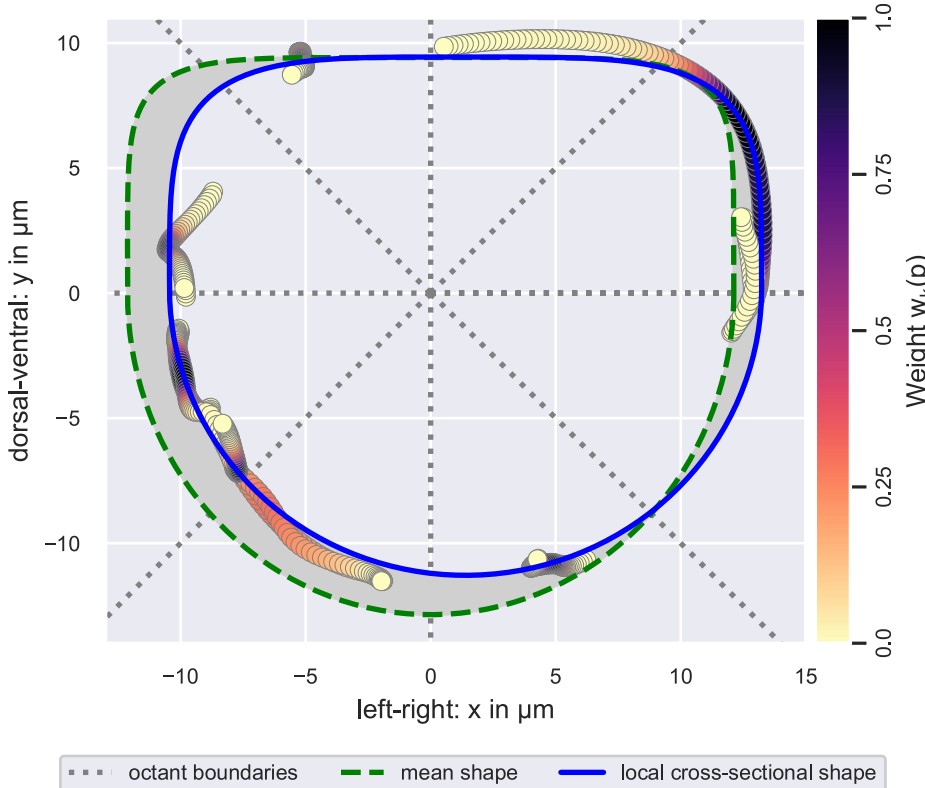

**Fig 4. Estimation of a local cross-sectional shape.** Visualized is the local cross-sectional shape estimated from points (circles) on EC contour splines that were projected onto the cross-sectional (*xy*-)plane. Data points are colored according to their weight during the estimation, i.e., their distance along the *z*-axis from the local cross-sectional plane (see color bar). Note that data points with a weight of zero are omitted in the figure. The employed weight function ensured that the estimation was based on at least $n_{\text{oct}}$ data points with non-zero weight per plane octant. Initial parameter values for the local cross-sectional shape were provided by the mean shape. During estimation, the deviation of the local cross-sectional shape from the mean shape, i.e., the area of the gray region relative to the area within the green dashed curve, was bound from above by λ. The illustrated cross-sectional shape was estimated using $n_{\text{oct}} = 30$ and λ = 20%.

sectional shapes from cell junctions only, we considered all points on cell contour splines during each estimation: To evaluate the goodness of fit of the estimate $\varphi$ with parameters $\theta$ on the cross-sectional plane at $z_k$, we first computed the projections $\Pi_\theta^{\text{shape}}([p]_{xy})$ of the *xy*-coordinates of all cell contour spline points $p \in \text{EC}_{\text{transf}}$ onto $\varphi$. We then defined the goodness of fit as a weighted sum of projection distances (see Eq (17)). To take into account that points closer to the cross-sectional plane carry more information on the local cross-sectional shape, we employed truncated Gaussian weights

$$w_k(p) := \begin{cases} \exp\left(-\dfrac{([p]_z - z_k)^2}{2(\omega_k)^2}\right); & [p]_z \in \left[z_k - Z_\omega \cdot \omega_k,\ z_k + Z_\omega \cdot \omega_k\right] \\ \\ 0 \end{cases} \qquad (15)$$

decaying with the distance of point $p$ to the cross-sectional plane at $z_k$. Here, $\omega_k > 0$ and $Z_\omega > 0$ are tuning parameters controlling the standard deviation and the distance of the truncation point in either direction from the mean value of the underlying Gaussian function, respectively.

The density of data points varied along the anterior-posterior axis, e.g., fewer data points were available at the annotation boundary than in the center of the anterior-posterior axis. Hence, we chose the width $\omega_k$ of the weight function in Eq (15) adaptive in space. To decide whether sufficient data points were available, we introduced the tuning parameter $n_{\text{oct}} \in \mathbb{N}$: We chose the minimal value for $\omega_k$ such that, when cell contour spline points $\text{EC}_{\text{transf}}$ were projected onto the $xy$-plane, at least $n_{\text{oct}}$ points $p \in \text{EC}_{\text{transf}}$ with non-zero weight $w_k(p)$ were located in each octant of the cross-sectional plane. The plane octants are illustrated in Fig 4.

In some data sets, i.e., embryos at either time point, increasing the minimal number of points per plane octant is not sufficient to estimate a physiologically plausible cross-sectional shape: For example, if many points with high weights are located in ventral octants, but only points with low weights are located in dorsal octants, unconstrained estimation can result in cross-sectional shapes that fit closely to the ventral points but deviate strongly from the dorsal points (see for example S10(D) Fig). To avoid this local overfitting, we additionally constrained the deviation $\text{dev}_{\text{cross}}(\varphi_k; \bar{\varphi})$ of the local cross-sectional shape $\varphi_k$ from the mean shape $\bar{\varphi}$ (see Eq (18)). We defined this deviation as the area of the symmetric difference of the two shapes, normalized by the mean shape's luminal area:

$$\text{dev}_{\text{cross}}(\varphi_k; \bar{\varphi}) := \frac{\text{area}(\varphi_k \triangle \bar{\varphi})}{\text{area}(\bar{\varphi})}, \tag{16}$$

where $\triangle$ is the symmetric difference of the interiors of two closed 2D curves (see gray region in Fig 4). To evaluate these symmetric differences, we approximated each curve by a polygon.

Eventually, we solved the following minimization problem for the estimation of the parameters $\theta_k$ of the cross-sectional shape at $z_k$:

$$\theta_k = \underset{\theta \in \mathbb{R}^7}{\text{argmin}} \sum_{p \in \text{EC}_{\text{transf}}} w_k(p) \cdot \left|\left| [p]_{xy} - \Pi_\theta^{\text{shape}}([p]_{xy}) \right|\right|_2 \tag{17}$$

$$\text{subject to} \quad \text{dev}_{\text{cross}}(\varphi_k; \bar{\varphi}) \leq \lambda, \tag{18}$$

where $\lambda \geq 0\%$ is a tuning parameter that controls the strength of the constraint.

**b) Smoothing of vessel surface.** In Eq (17), the estimation of the different cross-sectional shapes is decoupled from each other; they only depend on the same input data. This drastically reduces the dimensionality of the optimization problem. Also, all optimization problems can be solved in parallel, which greatly reduces computation time. As a result, the transition of the estimated cross-sectional shapes, however, is not smooth along the anterior-posterior axis. To smoothen these transitions, we employed a Gaussian filter on the parameters of the cross-sectional shapes. This Gaussian filter is controlled by two tuning parameters: the standard deviation $\sigma > 0$ of the underlying Gaussian function and $Z_\sigma > 0$ that regulates the distance of the Gaussian kernel function's truncation point in either direction from its mean value (comparable to $\omega$ and $Z_\omega$ in Eq (15)). We denote the cross-sectional shape at $z_k$ with smoothed parameters $\tilde{\theta}(z_k)$ as $\tilde{\varphi}_k(u) := \varphi(u; \tilde{\theta}(z_k))$. Further details on vessel surface smoothing are provided in Section 4 of S1 Appendix. To see an exemplary outcome of vessel surface smoothing, compare panels (iii) and (iv) of S10(E) Fig.

**c) Computation of grid.** To later enable triangulation of cell surfaces, we approximated the reconstructed vessel surface as a grid (also called organized/ordered point cloud in the literature). To this end, we computed $n_k$ equidistant points $\text{CS}_k$ on each cross-sectional shape $\tilde{\varphi}_k(u)$:

$$\text{CS}_k = (p_1, p_2, ..., p_{n_k}), \tag{19}$$

where points

$$p_j = \left( [\tilde{\varphi}_k(u_j)]_x, \ [\tilde{\varphi}_k(u_j)]_y, \ z_k \right); \quad j = 1, 2, \ldots, n_k \tag{20}$$

have parameter values

$$0 = u_1 < u_2 < \ldots < u_{n_k} < 2\pi. \tag{21}$$

To refer to the parameter value of point $p_j$ on the associated cross-sectional shape $\tilde{\varphi}_k$, we write $[p_j]_u := u_j$. To homogenize the density of points computed on cross-sectional shapes with different circumferences, we scaled the number $n_k$ of points computed per cross-sectional shape $\tilde{\varphi}_k$ with its circumference:

$$n_k = \left\lfloor n_{\text{cross,rel}} \cdot \int_0^{2\pi} ||\tilde{\varphi}_k'(u)||_2 \, du \right\rfloor, \tag{22}$$

where $n_{\text{cross,rel}}$ is a tuning parameter that controls the relative number of points computed per unit length (here: μm) for all cross-sections.

Analogous to the EC contours, we denote the successor/predecessor of a point $p \in \text{CS}_k$ using $p^+$ and $p^-$, respectively:

$$p_1^+ = p_2; \qquad p_2^+ = p_3; \qquad \cdots; \qquad p_{n_k}^+ = p_1 \tag{23}$$

$$p_{n_k}^- = p_{n_k-1}; \qquad p_{n_k-1}^- = p_{n_k-2}; \qquad \cdots; \qquad p_1^- = p_{n_k} \tag{24}$$

Throughout the text, it is typically clear from the context, whether $p^+$ or $p^-$ refers to neighbors within EC contours or within cross-sections. If this is not the case, clarifications are provided. Concatenating the points on all estimated cross-sections ($k = 1, 2, \ldots, M$), an ordered sequence

$$\text{CS} = \text{CS}_1 | \text{CS}_2 | \ldots | \text{CS}_M \tag{25}$$

of length $\sum_{k=1}^{M} n_k$ is obtained. We provide details on the computation of equidistant points on superelliptic shapes in Section 5.2 of S1 Appendix.

### Step 3: Endothelial cell surface triangulation

**a) Projection of endothelial cell contours onto vessel surface.** To define the EC surface boundaries on the estimated vessel surface, we projected each cell contour spline onto the grid approximation of the vessel surface. Whenever a spline $s_{\text{transf},i}$ intersected with a cross-sectional plane of the vessel at $z_k$, the intersection point was determined. The projection of this intersection point onto the vessel surface was then approximated by the point's nearest neighbor among the equidistant points $\text{CS}_k$ on the local cross-sectional shape. We denote the points of the projected spline contour of cell $i$ as $\text{EC}_{\text{proj},i}$. Details on the contour projection are provided in Section 6 of S1 Appendix. To see an exemplary outcome of cell contour projection, compare S8(C) and S8(D) Fig.

**b) Triangulation within projected endothelial cell contours.** To obtain a smooth surface for each EC, we triangulated each cell individually ensuring that all edges of the projected cell contour were part of the triangulation. The triangulation was performed in-between each pair of neighboring cross-sections (see Fig 5): First, we collected all edges between subsequent points $p, q \in \text{EC}_{\text{proj},i}$ that connected cross-sections $k$ and $(k + 1)$, i.e., $p \in \text{CS}_k$ and $q \in \text{CS}_{k+1}$. If the edges connecting cross-sections $k$ and $(k + 1)$ were $(p_1, q_1)$ and $(p_2, q_2)$, we connected the

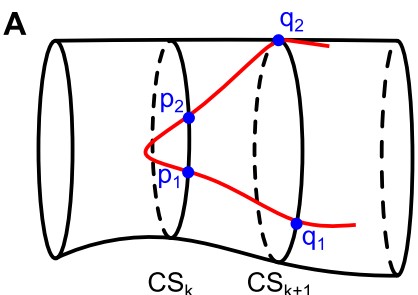 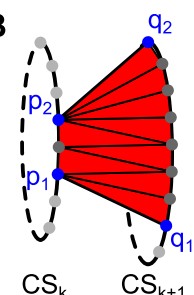

**Fig 5. Triangulation within a projected cell contour. (A)** In red, part of the projected cell contour on the smoothed vessel cross-sections is shown. To triangulate the part of the cell surface between cross-sections $k$ and $(k + 1)$, the edges $(p_1, q_1)$ and $(p_2, q_2)$ of the projected cell contour between these two cross-sections are identified. **(B)** A triangulation connects the cross-sectional path between $p_1$ and $p_2$ and the cross-sectional path between $q_1$ and $q_2$ within the cell surface. This triangulation is performed for each pair of neighboring vessel cross-sections that contains parts of the cell surface.

cross-sectional paths within the cell surface between $p_1$ and $p_2$ and between $q_1$ and $q_2$ via triangles. Our custom triangulation algorithm is explained in detail in Section 7 of S1 Appendix.

## Step 4: Analysis of vessel geometry and endothelial cell morphology

**a) Quantification of cross-sectional geometry.** We characterized cross-sectional geometry by two geometric measures:

1. The luminal area of the cross-sectional shape at $z_k$ was approximated by the area of the polygon enclosed by cross-sectional points $CS_k$:

$$\text{area}_{\text{lum},k} = \text{area}(CS_k) \in (0, \infty). \tag{26}$$

2. We defined the luminal diameter of the cross-sectional shape at $z_k$ as the maximal distance between any two points $p, q \in CS_k$ on the vessel cross-section:

$$\text{diam}_{\text{lum},k} := \max_{p,q \in CS_k} ||p - q||_2 \in (0, \infty). \tag{27}$$

**b) Quantification of endothelial cell surface morphology in 3D.** We denote by $T_i$ the set of triangles representing the surface mesh of endothelial cell $i$. Each triangle $\Delta \in T_i$ consists of three points in 3D space:

$$\Delta = \{p_1, p_2, p_3\} \subset \mathbb{R}^3. \tag{28}$$

Based on this mesh, we characterized each endothelial cell surface by several morphometric measures:

1. The surface area of cell $i$ was computed as the sum of the areas of its mesh's triangles:

$$\text{area}_i = \sum_{\Delta \in T_i} \text{area}(\Delta) \in (0, \infty), \tag{29}$$

where $\text{area}(\Delta)$ is the area of triangle $\Delta$.

2. The perimeter of cell $i$ was computed as the sum of the lengths of the projected cell contour edges:

$$\text{perim}_i = \sum_{p \in \text{EC}_{\text{proj},i}} ||p^+ - p||_2 \in (0, \infty). \tag{30}$$

3. The compactness of cell $i$ is defined as the ratio of the cell's surface area to the area of a circle with the same perimeter:

$$\text{compact}_i = \frac{4\pi\,\text{area}_i}{\text{perim}_i^2} \in (0, 1]. \tag{31}$$

A circle has compactness 1, lower values correspond to elongated cells or cells with protrusions.

4. To quantify elongation of a cell surface in the direction of flow, we first defined a cell surface bounding box. This box was chosen such that it covered the cell's extension within the cross-sectional plane and the cell's length in the direction of flow. An example of this box is illustrated in Fig 6A.
We defined elongation as the ratio of the bounding box's extension in the direction of flow ($z$) to its extension perpendicular to the direction of flow:

$$\text{elong}_i := \frac{\text{extension of box in } z\text{-direction}}{\text{extension of box within cross-sectional plane}} \in (0, \infty). \tag{32}$$

If $\text{elong}_i > 1$, cell $i$ is elongated in the direction of flow; the higher the value, the more pronounced the elongation in the direction of flow. If $\text{elong}_i \leq 1$, cell $i$ is not elongated in the direction of flow; the lower the value, the more pronounced the elongation perpendicular to the direction of flow. We provide details on the computation of the bounding box and cell elongation in Section 8 of S1 Appendix.

Finally, to quantify within-embryo variability in geometric or morphometric measures, we computed the quartile coefficient of dispersion (QCD):

$$\text{QCD} = \frac{Q_3 - Q_1}{Q_3 + Q_1} \in [0\%, 100\%], \tag{33}$$

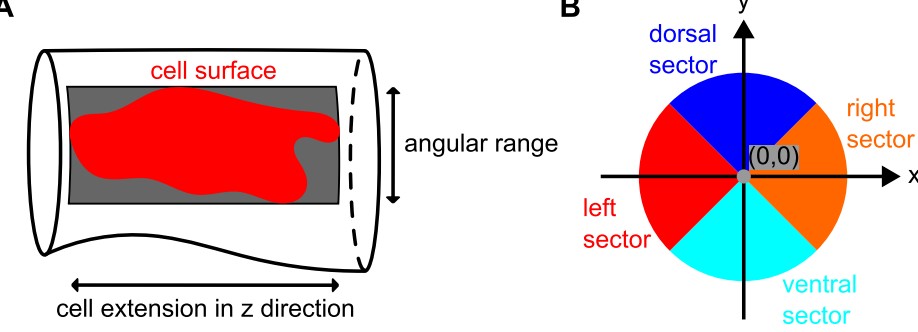

**Fig 6. Details on characteristics used during morphological analysis. (A)** The bounding box of a cell surface spans the entire angular range covered by the cell on the vessel surface along its length in $z$-direction. **(B)** Cross-sectional sectors for classification of cell location.

where $Q_1$ and $Q_3$ are the first and third quartiles of related measurements, e.g., EC surface areas, in an embryo at a single time point.

**c) Classification of endothelial cells.** We investigated whether ECs in different locations within the DA's cross-sections differed in their morphological characteristics. For this, we divided the entire DA surface into dorsal, ventral, left and right sectors (see Fig 6B). If a cell's surface area lying in the dorsal sector was greater than 50% of its total surface area, we classified the cell as a dorsal cell; ventral cells were analogously classified. We did not distinguish between left and right cells. Thus, if a cell's surface area lying in *either* the left *or* right sector was greater than 50% of its total surface area, we classified the cell as a left/right cell. For details on the cell classification, see Section 9.1–9.4 of S1 Appendix.

## Goodness of fit

To study the impact of contour preprocessing and vessel surface reconstruction on the level of individual cell contours, we quantified the distances between the manually annotated EC contours $EC_{anno}$ and their corresponding projected cell contours $EC_{proj}$. To this end, we employed an outlier-insensitive and symmetric distance measure between pairs of contours.

Consider two contours $A = (p_1, p_2, \ldots, p_{n_A})$ and $B = (q_1, q_2, \ldots, q_{n_B})$ of points $p_j \in \mathbb{R}^3$ with $j = 1, 2, \ldots, n_A$ and $q_{j'} \in \mathbb{R}^3$ with $j' = 1, 2, \ldots, n_B$. We first defined the distance $d_B(p)$ of a point $p \in A$ to contour $B$ as the minimal orthogonal distance to line segments $(q, q^+)$ with $q$, $q^+ \in B$ (see also Fig 7):

$$d_B(p) := \min_{q \in B} \left\| p - \Pi_{q,q^+}^{\text{edge}}(p) \right\|_2, \tag{34}$$

where $\Pi_{q,q^+}^{\text{edge}}(p)$ is the orthogonal projection of point $p$ onto the line segment $(q, q^+)$ (see Eq (30) in S1 Appendix).

Next, we defined the asymmetric distance $d_B(A)$ from contour $A$ to contour $B$ as the mean of the distances of points $p \in A$ to contour $B$:

$$d_B(A) := \text{mean}\left( \left\{ d_B(p) : p \in A \right\} \right). \tag{35}$$

While using the median instead of the mean in Eq (35) would make the distance measure more robust to single point outliers, we found that employing the mean made the distance measure more robust when two contours resembled each other along most of their length but differed substantially along an entire segment of the contour. As the distances $d_B(A)$, $d_A(B)$ depend on the point densities of contours $A$, $B$, respectively, we linearly interpolated additional points on contours $A$ and $B$ prior to distance computations.

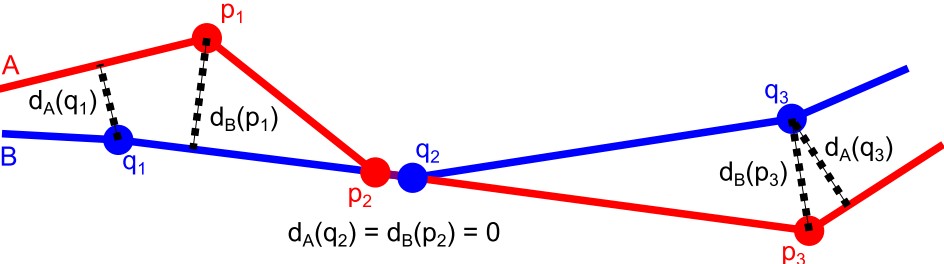

**Fig 7. Point distances to contours.** Distances (dotted lines) of points $p_1$, $p_2$ and $p_3$ on contour A and points $q_1$, $q_2$ and $q_3$ on contour B to the other contour (see also Eq (34)).

We finally defined the symmetric distance $d(A, B)$ between contours $A$ and $B$ as the maximum of the two asymmetric distances $d_B(A)$ and $d_A(B)$:

$$d(A, B) := \max\left(\left\{ d_B(A), d_A(B) \right\}\right). \tag{36}$$

When the distance between a manually annotated EC contour and its projection onto the vessel surface is larger than the uncertainty stemming from the manual annotation process, this may indicate that (i) the chosen tuning parameter values for vessel surface reconstruction were not suitable (explained below), (ii) errors were introduced during manual annotation of the contour or (iii) the EC bends outwards from the DA as part of an intersegmental vessel.

### Criteria for choice of tuning parameters

Our novel mathematical approach is fine-tuned by 13 parameters (see also S2 Table). These tuning parameters can be broadly categorized as (1) controlling the overall *precision* of cell contour description or vessel surface reconstruction, (2) capturing the *complexity* of the underlying experimental data and (3) directly controlling the *local quality of estimated* vessel cross-sections. During preprocessing of manually annotated EC contours and vessel surface reconstruction, we applied the *same* tuning parameter values to all wild-type and Endoglin-deficient embryos at both 48 hpf and 72 hpf.

In short, tuning parameters controlling contour preprocessing were identified by visual inspection. Suitable values for the *data-complexity* parameters $r$ and $\Delta h$, which control EC contour enrichment by neighbors, ensure that the combined information of neighboring cells on a cell-cell contact is integrated into each of the cells' contours. Their magnitude depends only on the cell contours' complexity, i.e., the cells' deviation from an ellipse/hexagon due to irregular boundaries and protrusions. Values chosen for the *precision* parameters $n_{\text{interp,rel}}$, $\epsilon_{\text{spline}}$ and $n_{\text{spline,rel}}$, which control the approximation of EC contours by equidistant points on splines, should ensure that each EC contour is smoothed but still closely resembles its manual annotation. For further details, see Section 10.1 in S1 Appendix.

To precisely reconstruct local vessel geometry, we chose high values for the *precision* parameters $M_{\text{rel}}$, $n_{\text{poly}}$ and $n_{\text{cross,rel}}$ that control the number of estimated cross-sections along the vessel axis and the number of points computed on these cross-sections. *Local-estimation-quality* parameters $Z_\omega$ and $Z_\sigma$ controlling the truncation of the weight and smoothing functions were fixed to default values. The presented vessel surface reconstruction method balances the locality of estimated cross-sectional shapes, the physiological plausibility of these shapes and the overall smoothness of the vessel surface. These trade-offs are controlled by the three *local-estimation-quality* parameters $n_{\text{oct}}$, $\lambda$ and $\sigma$. For different combinations of the parameters' values, we measured goodness of fit by computing the distances between the manually annotated EC contours $\text{EC}_{\text{anno}}$ and their projections $\text{EC}_{\text{proj}}$ onto the vessel surface using Eq (36). Additionally, we visually inspected the physiological plausibility of the estimated shapes and the smoothness of the resulting vessel surface. In large angular sectors of the cross-sectional plane that only contained points with low weights during the shape's estimation, a physiologically plausible cross-sectional shape would either locally resemble the mean shape or closely match those points with low weights. For further details, see Section 10.2 in S1 Appendix.

As the shapes of estimated vessel cross-sections are directly controlled by the *local-estimation-quality* parameters $n_{\text{oct}}$, $\lambda$ and $\sigma$, we next tested whether the parameters' values were robust by performing a sensitivity analysis: We investigated the effect of these parameters on vessel geometry and EC morphology by re-estimating each vessel surface with altered value of either $n_{\text{oct}}$, $\lambda$ or $\sigma$, while fixing the other two values. We then quantified the deviation

$\mathrm{dev_{cross}}(\varphi^*;\varphi)$ of each re-estimated cross-sectional shape $\varphi^*$ from its corresponding reference shape $\varphi$ using Eq (16). Additionally, we quantified the relative deviation $\mathrm{dev_{rel}}(\mathrm{measure}_i^*; \mathrm{measure}_i)$ of each altered morphometric measurement $\mathrm{measure}_i^*$, computed on a re-estimated vessel surface, from its corresponding reference value $\mathrm{measure}_i$ as

$$\mathrm{dev_{rel}}(\mathrm{measure}_i^*; \mathrm{measure}_i) = \frac{\mathrm{measure}_i^* - \mathrm{measure}_i}{\mathrm{measure}_i^*}. \tag{37}$$

For example, if $\mathrm{dev_{rel}}(\mathrm{perim}_i^*; \mathrm{perim}_i^*) = -10\%$, the EC perimeter $\mathrm{perim}_i^*$, computed on a re-estimated vessel surface, is 10% lower than its reference value $\mathrm{perim}_i$. We computed relative deviations of altered geometric measurements in the same manner.

## Validation of mathematical approach

To verify that our mathematical approach enables accurate reconstruction of DA cross-sections using only information from EC contours, we compared the vessel surface estimated from EC contours in the DA of two wild-type embryos at 72 hpf to manually annotated angiogram slices of the same DAs (validation data in Table 1). During surface reconstruction, we used the same parameter values as in the primary analysis. We then compared the estimated cross-sectional geometry to the geometry of the annotated angiogram slices. As the planes containing annotated angiogram slices depended on the position of the embryos within the mounting medium during imaging and did not match cross-sectional planes in our estimated coordinate system, we sliced vessel surfaces estimated from cell contours with planes containing the angiogram slices. To then quantify differences in the (polygonal) shapes of vessel surface slice $S$ and corresponding angiogram slice $S_{\mathrm{ref}}$, we computed their relative deviation $\mathrm{dev_{cross}}(S; S_{\mathrm{ref}})$ using Eq (16). Additionally, we quantified the relative deviation $\mathrm{dev_{rel}}(\mathrm{area}(S); \mathrm{area}(S_{\mathrm{ref}}))$ of their luminal areas using Eq (37).

Since prior methods had employed ellipses instead of superellipses to describe DA cross-sections [3, 15], we also investigated the impact of the employed cross-sectional shape model on the estimated vessel geometry. To this end, we re-estimated the vessel surfaces using either circular or elliptic cross-sectional shapes. To then quantify the deviation of the estimated circular or elliptic cross-sectional geometries from the geometry of the annotated angiogram slices, we again applied Eqs (16) and (37).

## Implementation

We implemented the reconstruction of zebrafish DA geometry and EC surface morphology from manually annotated EC contours in a python package called **CO**ntour-based **VE**ssel surface **R**econstruction (COVER). This package uses python 3.12, numpy [21], pandas [22] and scipy [23]. From scipy, we employed routines for minimization, root finding, quadrature, splines, nearest neighbor searches via KD trees and correlation analysis. We computed symmetric differences and polygonal areas with shapely [24]. Principal component analysis was performed using scikit-learn [25]. Furthermore, we employed pyvista [26] for visualization of triangular meshes, mesh clipping and mesh slicing. During method development and whenever inspecting intermediate results, we produced interactive plots with plotly [27]. Most plots that are referenced in the results section of this article were created using matplotlib [28] and seaborn [29].

The analysis' underlying microscopy images including the manually annotated EC contours can be inspected using the free Imaris Viewer software (Bitplane). To allow comparison of the manually annotated EC contours against their respective enriched contours, smoothing splines and their projections onto the estimated vessel surfaces, we additionally inserted these

intermediate results into the Imaris files. The computed EC surface meshes were exported in Visualization Toolkit file formats [30]. These files can be inspected without any programming experience using the open-source software Paraview [31] (see instructions in S1 Tutorial).

## Results

Most results were computed using the analysis data (see Table 1); evaluations using the validation data are explicitly indicated.

### Robust choice of tuning parameter values

We chose tuning parameter values in accordance with the criteria summarized above. A summary of the parameters' functions and their chosen values is provided in S2 Table. Tuning parameter values used during contour preprocessing visibly reduced overlaps and gaps between neighboring cell contours, which we initially observed for the manually annotated cell contours. After vessel surface reconstruction, we computed the distances between the manually annotated EC contours $EC_{anno}$ and their projections $EC_{proj}$ onto the vessel surface using Eq (36). For the chosen tuning parameter values, we found that the resulting median contour distances were 45% lower than those that were obtained when the vessel segment was instead described by a constant cross-sectional shape along its entire length (see S10(B) and S10(C) Fig). Additionally, we confirmed by visual inspection that the estimated vessel cross-sections had physiologically plausible shapes and together formed a smooth vessel surface (see exemplary illustrations in S10(D) and S10(E) Fig). Although we observed that tuning parameters $n_{oct}$, $\lambda$ and $\sigma$ critically determined the shapes of estimated and smoothed vessel cross-sections (see exemplary illustrations in S10(D) and S10(E) Fig), chosen values for these parameters were robust: when halving or doubling either $n_{oct}$, $\lambda$ or $\sigma$, the majority of estimated vessel cross-sections had a relative deviation of less than 5% (see S3 Table) and no morphometric measurement differed by more than 10% (see S4 Table).

### Accurate projection of endothelial cell contours

For the chosen tuning parameter values, we analyzed whether the distances between the manually annotated cell contours $EC_{anno}$ and their projections $EC_{proj}$ onto the estimated vessel surface were sufficiently low to allow precise analysis of EC morphology. For this, we compared these distances with the uncertainty resulting from the manual annotation process: The distances between the manually annotated cell contours $EC_{anno}$ (analysis data) and their projections $EC_{proj}$ onto the estimated vessel surfaces had a median/maximum of 0.474 μm/1.36 μm, respectively. These distances were comparable to the median/maximal distances of 0.535 μm/ 0.742 μm found between the two annotation versions of the same cell contours in two wild-type embryos at 72 hpf (validation data; see Fig 8).

We inspected ECs with high projection distances (analysis data). Here, we found that these were cells that partially bend outwards from the DA as part of an intersegmental vessel or cells at the annotation boundary where, locally, an insufficient number of data points was available to accurately estimate local cross-sectional shapes. To ensure that we only quantified morphology in ECs whose contours were well represented on the estimated vessel surface, we excluded those 22 out of 296 cells (from the analysis data) that had projection distances greater than the maximal annotation uncertainty of 0.742 μm in the following morphological analysis.

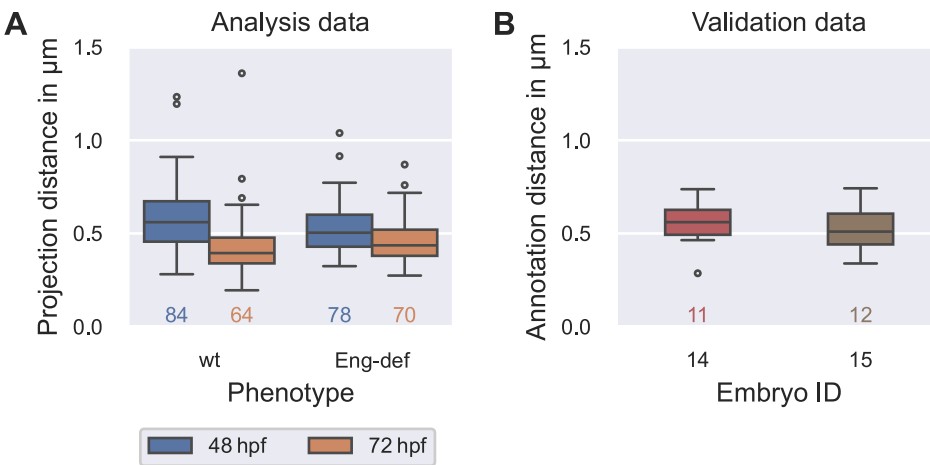

**Fig 8. Comparability of errors introduced by vessel surface reconstruction and annotation uncertainty. (A)** Distribution of the distances between manually annotated endothelial cell contours $EC_{anno}$ (analysis data) and their projections $EC_{proj}$ onto the estimated vessel surface (see Eq (36)), stratified by phenotype and time. **(B)** Distribution of distances (see Eq (36)) between first and second annotation of endothelial cells in two wild-type embryos at 72 hpf (validation data). The numbers below each box plot denote the corresponding numbers of cells.

## Accurate estimation of vessel cross-sections from cell contours

After confirming that our method was precise enough for the computation of EC morphology, we evaluated whether the estimation of vessel cross-sections using only information from EC contours resulted in an accurate description of DA geometry. For this, we compared vessel surfaces estimated from cell contours of two wild-type embryos at 72 hpf to angiogram slices from the same DAs (validation data). Pooling both embryos, we found that 92%-93% of vessel surface slices had luminal areas differing by at most 20% from the areas of angiogram slices when using either circular, elliptic or superelliptic cross-sections. For a more detailed analysis of luminal area in each of the two embryos, see S5 Table.

We further found that the *shapes* of estimated vessel cross-sections resembled the shapes of angiogram slices well for each tested cross-sectional shape. Ellipses and superellipses, however, were slightly more accurate in describing angiogram slices than circles: when pooling both embryos, the median deviation of vessel surface slices from the angiogram slices was 17%/14%/15% when using circular/elliptic/superelliptic cross-sections, respectively. For a more detailed analysis of luminal shape in each of the two embryos, see S11(A) Fig.

Using the same data, we then investigated how the density of annotated cell contours affected the approximation quality of the estimated (superelliptic) vessel cross-sections from the angiogram slices. Pooling both embryos, the median deviation of the estimated cross-sections from the angiogram slices was 61% higher for angiogram slices where only one annotated EC was located within a distance of 0.5 μm in comparison to slices with more nearby cells. For a more detailed analysis of the effect of annotation density on the deviation in luminal shape in the individual embryos, see S11(B) Fig. The effect of annotation density on the accuracy of cross-sectional shapes motivated us to exclude all cross-sections which had only one annotated cell within a distance of 0.5 μm to the cross-sectional plane (3697 out of 25 111 cross-sections) in the subsequent analysis of DA geometry (from the analysis data).

Subsequently, we applied our novel mathematical approach for an exploratory analysis of DA geometry and EC morphology in wild-type and Endoglin-deficient zebrafish embryos.

## High between-embryo but low within-embryo variability in luminal area

First, we quantified cross-sectional geometry in the DA of wild-type and Endoglin-deficient embryos at 48 hpf and 72 hpf (see Fig 9). We found that median luminal area in the pooled Endoglin-deficient embryos was 34% smaller at 48 hpf in comparison to the pooled wild-types, but 150% larger at 72 hpf. Comparing the individual embryos, we found that biological between-embryo variability in luminal area was remarkably high in both phenotypes at each time point: the maximal median luminal area over all wild-type embryos was 90%/49% higher than the corresponding minimal median luminal area at 48 hpf/72 hpf, respectively (compare Fig 9A). Among Endoglin-deficient embryos, we found that the maximal median luminal area was 71% higher than the minimum at 48 hpf and 57% higher at 72 hpf (compare Fig 9B). In contrast to the high between-embryo variability, biological *within*-embryo variability in luminal area was low in the analyzed short segments of the DA: the maximal QCD (see Eq (33)) over all embryos and time points was only 11%.

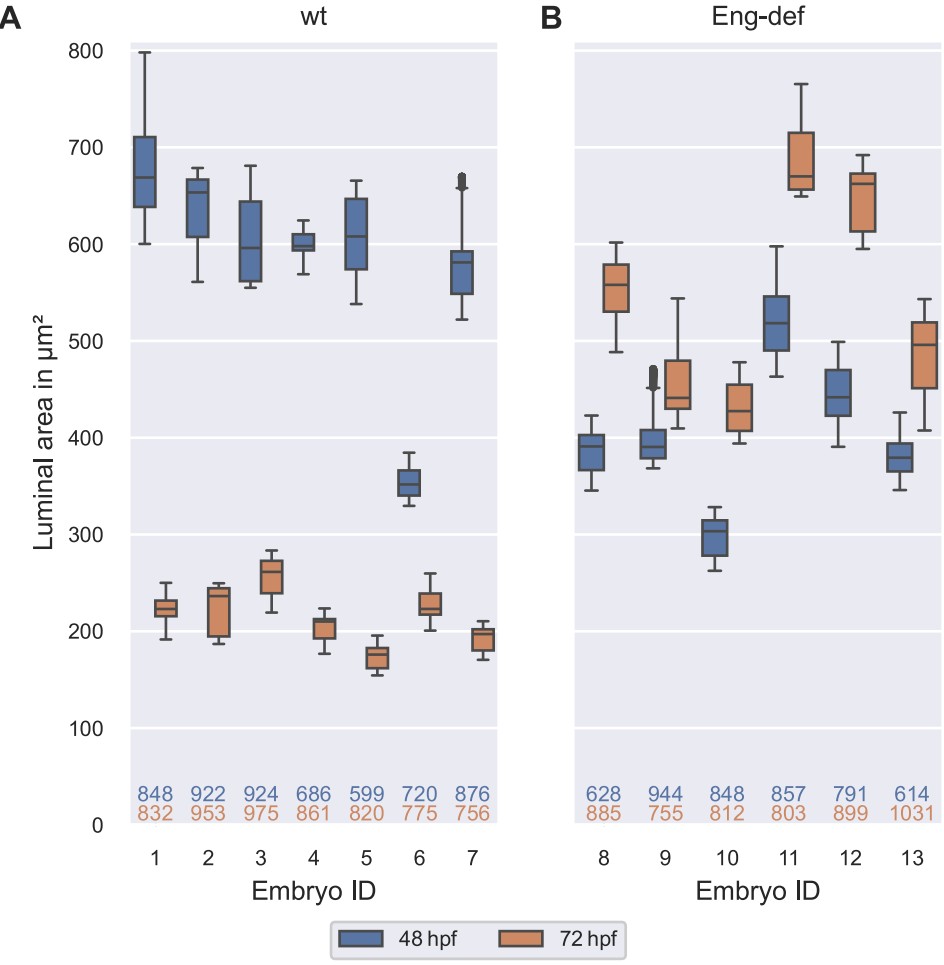

**Fig 9. High between-embryo but low within-embryo variability in luminal area.** Distribution of luminal areas of estimated DA cross-sections, stratified by embryo and time, for wild-type embryos (**A**) and for Endoglin-deficient embryos (**B**). Analysis based on cross-sections with at least two annotated cells present within a distance of 0.5 μm to the cross-sectional plane. The numbers below the box plots denote the corresponding numbers of vessel cross-sections. The top number corresponds to 48 hpf and the bottom number to 72 hpf.

When inspecting each embryo individually, we found that in each wild-type embryo, median luminal area decreased between 48 hpf and 72 hpf (see Fig 9A), while it *increased* in each Endoglin-deficient embryo (see Fig 9B). Relative changes in median luminal area between 48 hpf and 72 hpf however varied per embryo. For example, wild-type embryo 6, which already had the lowest median luminal area at 48 hpf, had a much smaller relative decrease in median luminal area of 37% in comparison to the other wild-type embryos where the relative decreases ranged from 56%-71%. Changes in luminal area also varied between Endoglin-deficient embryos. Here, increases in median luminal area ranged from 13%-50%. In summary, we found high between-embryo but *low within-*embryo variability in luminal area in short DA segments of both wild-type and Endoglin-deficient embryos.

## High between-embryo and high within-embryo variability in endothelial cell morphology

Following our evaluation of cross-sectional geometry, we quantified EC morphology in the same DAs of wild-type and Endoglin-deficient embryos at 48 hpf and 72 hpf (see Fig 10). We found that median EC surface area in the pooled Endoglin-deficient embryos was 21% smaller at 48 hpf in comparison to the pooled wild-types, but 40% *larger* at 72 hpf. Inspecting changes in EC surface area between 48 hpf and 72 hpf, we observed opposing trends in the two phenotypes: Among the wild-type embryos, embryos 1 and 3 were the only embryos where median cell surface area increased (by 43% and 6.6%, respectively; see Fig 10A). In the remaining five wild-types, *decreases* in median cell surface area ranged from 3.7% to 34%. In contrast, median cell surface area substantially *increased* in almost all Endoglin-deficient embryos; these increases ranged from 43% to 170% (see Fig 10B). Embryo 13 was the only Endoglin-deficient embryo where median cell surface area slightly *decreased* between 48 hpf and 72 hpf (by 17%).

Although we observed clear differences in EC surface area between the two phenotypes, perimeters were comparable! Median EC perimeter in the pooled Endoglin-deficient embryos differed by maximally 11% from that of the pooled wild-type embryos at each time point. When inspecting the individual embryos, we found that median EC perimeters increased in almost all embryos between 48 hpf and 72 hpf; the only *decrease* (by 4.0%) occurred in wild-type embryo 7 (see Fig 10C and 10D). Notably, increases in median perimeter were greater in Endoglin-deficient embryos: Excluding embryo 13 where median perimeter only increased by 1.9%, median perimeter increases ranged from 34% to 85% in the remaining Endoglin-deficient embryos, but only from 7.0% to 21% in wild-types.

Compactness of ECs was also similar in the two phenotypes: median compactness was 2.4% smaller in the pooled Endoglin-deficient embryos than in the pooled wild-types at 48 hpf but 7.4% larger at 72 hpf. When comparing individual changes in compactness between 48 hpf and 72 hpf, we found that median compactness decreased in each embryo (see Fig 10E and 10F). Between-embryo variability in this decrease was however higher in Endoglin-deficient embryos: Here, decreases in median compactness ranged from 4.5% to 31%, while decreases in wild-type embryos only ranged from 18% to 30%.

In comparison to the pooled wild-types, median EC elongation was 19% higher in the pooled set of Endoglin-deficient embryos at 48 hpf, but 28% *lower* at 72 hpf. Notably, median elongation increased in each embryo between 48 hpf and 72 hpf (see Fig 10G and 10H). These increases were however less pronounced in Endoglin-deficient embryos where they ranged from 14% to 48%, while increases in wild-type embryos ranged from 74% to 130%.

To analyze whether there were systematic relationships between any of the computed morphometric measurements, we next performed a correlation analysis (see S12 Fig). When pooling all ECs, we found that cell surface area and perimeter showed a strong positive correlation

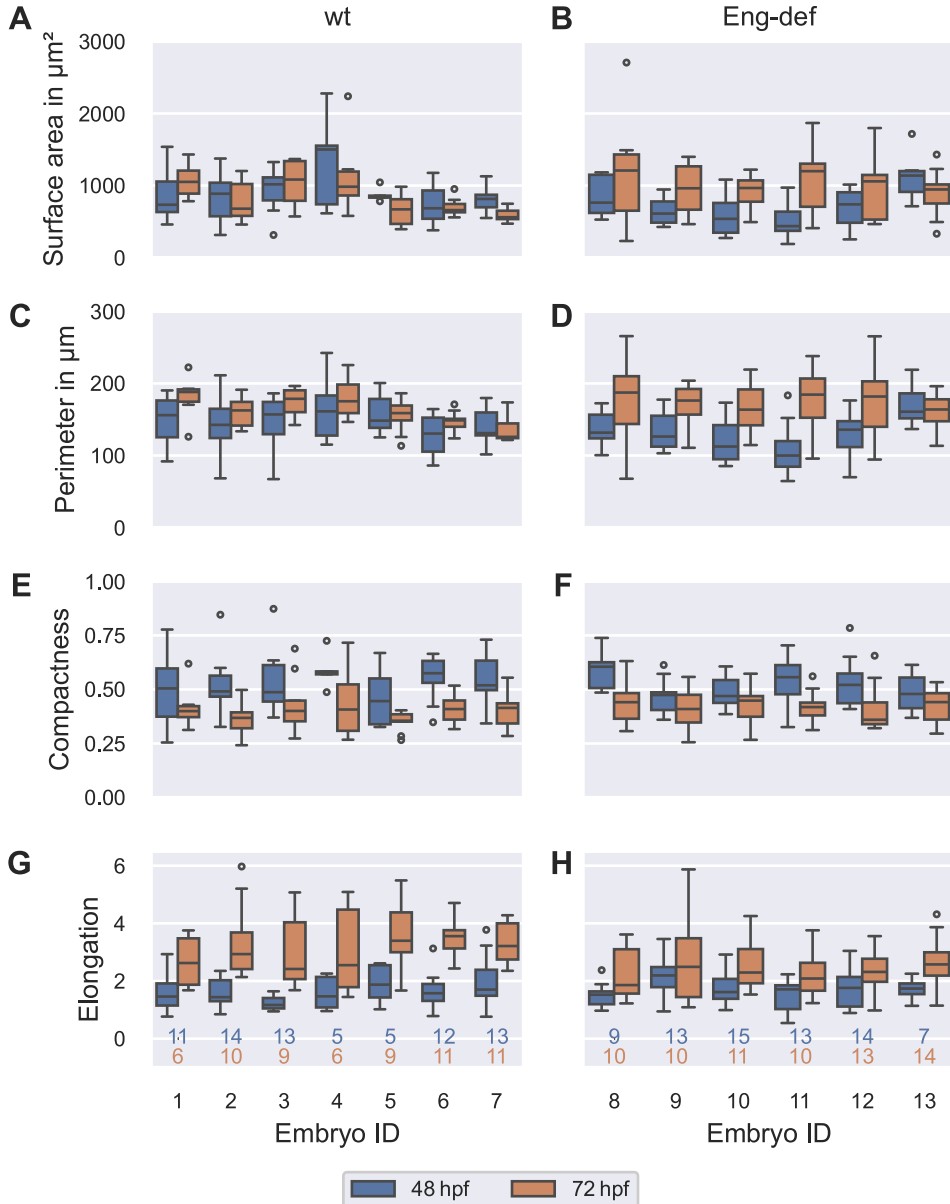

**Fig 10. High between-embryo and high within-embryo variability in endothelial cell morphology.** Distribution of morphometric measurements over all ECs, stratified by embryo and time, for wild-type embryos ((**A**), (**C**), (**E**) and (**G**)) and for Endoglin-deficient embryos ((**B**), (**D**), (**F**) and (**H**)). Analysis based on ECs with projection distances (onto the estimated vessel surfaces) comparable to or lower than the maximal annotation uncertainty of 0.742 μm. The numbers below the bottom box plots denote the corresponding numbers of cells for each column. The top number corresponds to 48 hpf and the bottom number to 72 hpf.

(Spearman rank correlation coefficient $r_s$ = 0.849). Also, when separating cells by phenotype and time, this correlation remained strong ($0.784 \leq r_s \leq 0.916$). This meant that ECs had few protrusions, otherwise large increases of perimeter could have been accompanied by small increases in surface area. Pooling all ECs, elongation and compactness had a moderately strong negative correlation ($r_s = -0.640$). When separating cells by phenotype and time, this correlation was slightly weaker ($-0.562 \leq r_s \leq -0.466$). In the absence of many protrusions, this

negative correlation implied that very elongated cells (with low compactness) had a tendency to be aligned in the direction of flow, i.e., they had a high value of our elongation measure.

Finally, we analyzed within-embryo variability in morphometric measures. Notably, we found that biological within-embryo variability in EC morphology was much higher than within-embryo variability in luminal area in the same DA (see S13 Fig). This was most evident in EC surface area and elongation: In 96%/100% of data sets, i.e., embryos at either time point, variability (QCD) in EC surface area/elongation was more than 50% higher than QCD of luminal area, respectively. These two morphometric measures also had systematically higher within-embryo variability than EC perimeter and compactness (see S14 Fig): In 85%/65% of data sets, QCD of EC surface area was more than 25% higher than QCD of perimeter/compactness, respectively. In 77% of data sets each, QCD of EC elongation was more than 25% higher than QCD of perimeter and compactness. In summary, we found high between-embryo and high within-embryo variability in EC morphology in both wild-type and Endoglin-deficient embryos. Among the morphometric measures, EC surface area and elongation in the direction of flow showed the greatest within-embryo variability.

## Dorsal-ventral asymmetry of endothelial cell morphology

We then analyzed whether stratification of cells by their location within the DA reduced the remaining high within-embryo variability in EC morphology, especially in EC surface area and elongation. For 253 out of 274 ECs, we were able to identify the cross-sectional sectors containing the largest part of the cells' surface area. This allowed us to study morphological differences between dorsal, ventral and left/right ECs in both wild-type and Endoglin-deficient embryos (see Fig 11).

In wild-type embryos, we found evidence of a dorsal-ventral asymmetry. Firstly, dorsal ECs were larger than ventral ECs: Median surface area of dorsal ECs was 32%/22% larger at 48 hpf/72 hpf, respectively (see Fig 11A). Differences in perimeter were less pronounced; median perimeter of dorsal ECs was 9.5% higher at 48 hpf but only 1.0% higher at 72 hpf (see Fig 11C). We further observed that dorsal ECs were more compact than ventral ECs but *less* elongated in the direction of flow: Median compactness of dorsal ECs was 9.6% larger at 48 hpf and 27% larger at 72 hpf (see Fig 11E). In contrast, median elongation was 47%/42% *lower* in dorsal ECs at 48 hpf/72 hpf, respectively (see Fig 11G). When investigating changes in these measures over time, we found that dorsal and ventral ECs differed most in surface area and perimeter: Between 48 hpf and 72 hpf, median surface area of dorsal ECs decreased by 14%, but ventral ECs only showed a decrease of 7.3% (see Fig 11A). While median perimeter only increased by 7.8% in dorsal ECs, its increase was 17% in ventral ECs (see Fig 11C).

Our analysis of EC morphology in Endoglin-deficient embryos also supported a dorsal-ventral asymmetry. We found that dorsal ECs were larger than ventral ECs at both time points, but this difference was much more pronounced at 72 hpf: In comparison to ventral ECs, median cell surface area of dorsal ECs was 23% larger at 48 hpf but 78% larger at 72 hpf (see Fig 11B). While median perimeter of dorsal ECs was 3.4% smaller at 48 hpf, it was 21% *larger* at 72 hpf (see Fig 11D). Again, we observed that dorsal ECs were more compact than ventral ECs but *less* elongated in the direction of flow: Median compactness of dorsal ECs was 11% larger at 48 hpf and 16% larger at 72 hpf (see Fig 11F). In contrast, median elongation was 37%/50% *lower* in dorsal ECs at 48 hpf/72 hpf, respectively (see Fig 11H). Dorsal and ventral ECs not only had large differences in cell surface area and elongation at both 48 hpf and 72 hpf but changes in these measures over time also differed in their magnitude: Between 48 hpf and 72 hpf, median surface area increased by 100% in dorsal ECs but only by 39% in ventral ECs

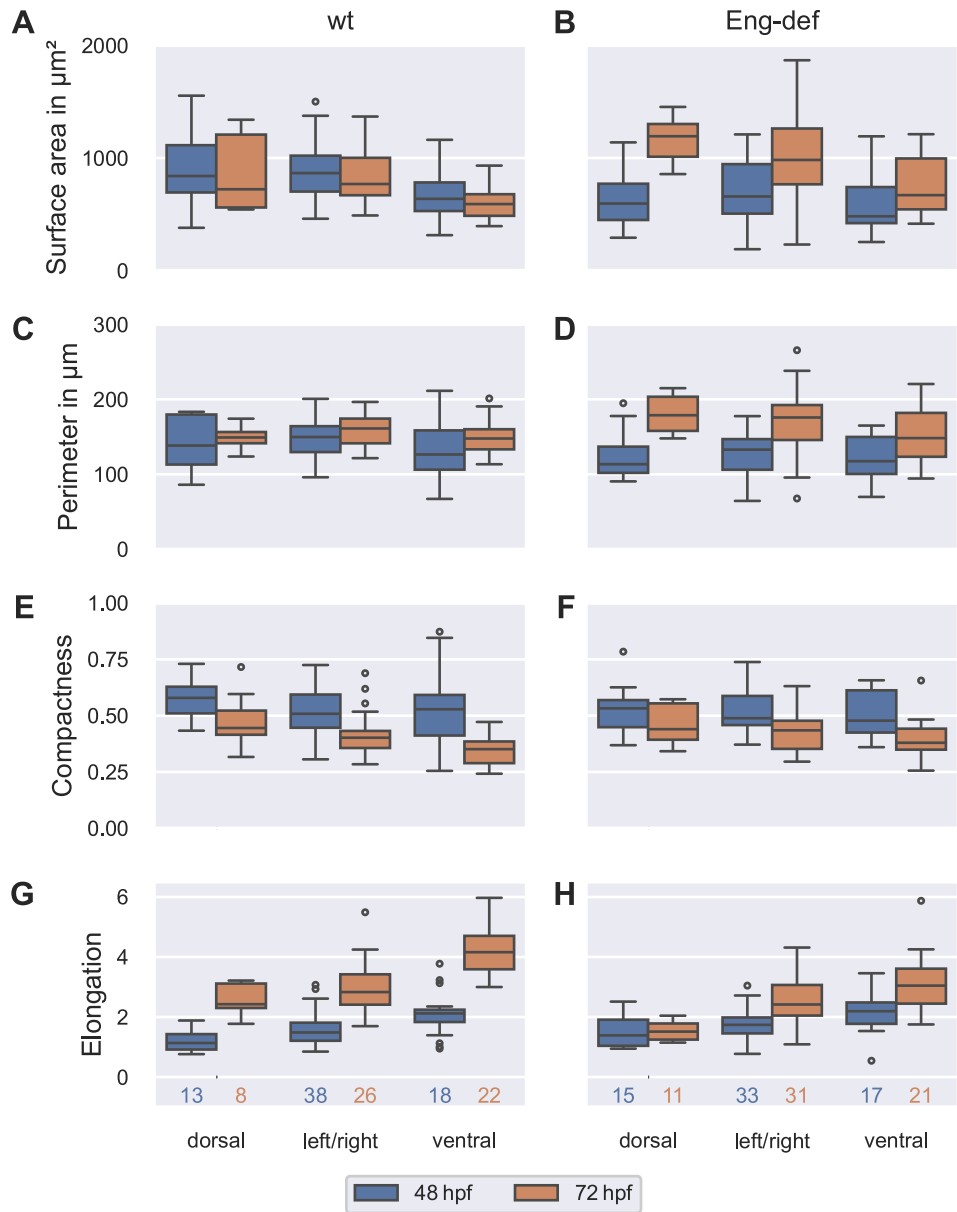

**Fig 11. Dorsal-ventral asymmetry of endothelial cell morphology.** Distribution of morphometric measurements over all ECs, stratified by location and time for wild-type embryos ((**A**), (**C**), (**E**) and (**G**)) and for Endoglin-deficient embryos ((**B**), (**D**), (**F**) and (**H**)). Analysis based on ECs with projection distances (onto the estimated vessel surfaces) comparable to or lower than the maximal annotation uncertainty of 0.742 μm. The numbers below the bottom box plots denote the corresponding numbers of cells for each column.

(see Fig 11B). Simultaneously, median elongation of dorsal ECs barely increased (by 9.0%), while median elongation of ventral ECs increased by 39% (see Fig 11H).

As differences between wild-type and Endoglin-deficient embryos were most pronounced in EC surface area and elongation, we finally compared surface area and elongation in dorsal and in ventral ECs between the two phenotypes. For each of the two morphometric measures, we found that dorsal ECs showed the greatest differences between the two phenotypes: At 48 hpf, median cell surface area of dorsal ECs was 29% smaller in Endoglin-deficient embryos

than in wild-types, while median cell surface area of ventral ECs was only 24% smaller in Endoglin-deficient embryos (see Fig 11A and 11B). At 72 hpf, differences between the two phenotypes were even more pronounced in dorsal ECs: here, median cell surface area of dorsal ECs was 65% larger in Endoglin-deficient embryos than in wild-types, while median cell surface area of ventral ECs was only 13% larger in Endoglin-deficient embryos. Comparing elongation between the two phenotypes, we found that median elongation of dorsal ECs was 22% higher in Endoglin-deficient embryos at 48 hpf (see Fig 11G and 11H). In contrast, median elongation of ventral ECs of Endoglin-deficient embryos was only 3.5% higher. At 72 hpf, both dorsal and ventral ECs showed greater differences between the two phenotypes: here, median elongation of dorsal ECs was 38% *lower* in Endoglin-deficient embryos than in wild-types, while median elongation of ventral ECs was only 27% lower.

Finally, we closely inspected those ECs that could not be classified (see S15 Fig). These cells contained cell surface areas in at least three of the dorsal, ventral, left and right sectors. Most of the cells had the largest surface area contribution in the dorsal sector (14 out of 21 ECs); no EC had the largest surface area contribution in the ventral sector. In each phenotype and at each time point, non-classifiable ECs had higher median cell sizes than dorsal, ventral and left/right ECs. Median compactness hardly differed between non-classifiable ECs and the classified ECs. Lastly, non-classifiable ECs had the lowest median elongation in the direction of flow. Taken together, the large size of the non-classifiable ECs, their low elongation in the direction of flow and predominant location in the dorsal sector further support a dorsal-ventral asymmetry of endothelial cell morphology.

In summary, we found that dorsal and ventral ECs in the DA of both wild-type and Endoglin-deficient embryos vastly differed in their morphology, especially in their surface areas and elongation in the direction of flow. Furthermore, dorsal ECs showed the most pronounced differences between wild-types and Endoglin-deficient embryos. We confirmed these findings using a finer classification (see Section 9.5 in S1 Appendix), where we distinguished cells that were exclusively located in a single cross-sectional sector from those that wrapped around a larger fraction of the DA (see S16 Fig).

Solely for the purpose of visualization of the identified morphological differences between dorsal and ventral ECs, we further simplified blood vessel geometry: We described vessel cross-sections by the (constant) mean shape along the entire length of the analyzed segments, thus neglecting potentially important variations in local geometry. This allowed us to unroll the vessel surface in a second step circumventing cartographic distortions (see S17 Fig). Note however that the hereby resulting increases in contour projection distance were substantial for some of the analyzed cells.

## Discussion

In this article, we developed a novel mathematical approach that enables reconstruction of the zebrafish DA's apico-luminal surface using only data from annotated EC contours and approximate directions of the vessel's anterior-posterior, dorsal-ventral and left-right axes; information from EC surface markers is not required. We found that errors on the EC contour level that were introduced by vessel surface reconstruction were comparable to the uncertainty stemming from the manual contour annotation process. Furthermore, we validated the accuracy of our EC contour-based vessel surface reconstruction method by comparing the luminal shapes of estimated vessel surfaces of wild-type embryos against angiography of the same DAs. Application of our mathematical approach enabled us to precisely and robustly quantify DA geometry and 3D EC surface morphology in wild-type and Endoglin-deficient zebrafish embryos at 48 hpf and 72 hpf.

To ensure a physiologically accurate description of the DA's surface, we developed a novel cross-sectional shape model that explicitly accounts for the vessel's dorsal flattening that we had observed in 3D microscopy images. We found that vessel surfaces estimated using the novel superelliptic shape model described angiography data of the DA similarly well as frequently employed ellipses. Compared with ellipses, however, our shape's dorsal-ventral asymmetry can not only accurately describe the DA's geometry but also aid in identification of a physiologically plausible coordinate system of the vessel from initial direction vectors, while neither the rotation angle nor the semi-axis lengths of an ellipse are informative with respect to the vessel's dorsal-ventral axis. Independent of the employed cross-sectional shape, an alternative way to identify the DA's dorsal-ventral axis is to leverage further spatial information, e.g., the positions of ECs of the ventrally located caudal vein as done in [15]. While both approaches result in physiologically meaningful coordinate systems that allow to compare DAs in different embryos, our method does not require any data from neighboring vessels.

A major challenge during vessel surface reconstruction was that information on the left and right side of the DA was imbalanced: Since embryos lay on their right side within the confocal microscope, their left side was further away from the light source and therefore prone to bleaching effects and signal scattering, resulting in a loss of resolution and fluorescence signal; this made EC contour annotation on the left more difficult. This was more prominent for embryos at 48 hpf, since their larger yolk sacks increased the distances of their DAs from the microscope. Here, our novel cross-sectional shape model again aided in the estimation process: Whenever cell contours on the DA's left side could not be annotated with confidence (and were thus omitted from the analysis), the shape's symmetry axis and employing constrained minimization helped to estimate physiological cross-sectional shapes from an incomplete set of local cell contours. Freely rotating ellipses cannot accomplish this.

By first estimating vessel cross-sections from EC contours and then projecting each contour onto the estimated cross-sectional shapes, our approach not only enabled us to describe variability in EC surface morphology but also explain it by the cells' location within the vessel cross-sectional plane. While the former is also possible when only estimating individual EC surfaces like in [3], the latter additionally requires the reconstruction of vessel geometry in a manner that is consistent with all ECs. Since we based the estimation of vessel cross-sections on the combined information of *all* EC contours, our approach can additionally compensate for annotation errors in single cell contours by contours of cells that belong to the same vessel cross-sections. The EC surfaces produced by our method are thus less sensitive to uncertainty stemming from the manual annotation process than EC surfaces whose estimation is only informed by single cell contours like in [3].

As an additional evaluation of the quality of our experimental data and the accuracy of our cell contour-based vessel surface reconstruction, we scanned the literature for data on DA geometry and morphology of its constituent ECs. While we found many studies measuring DA geometry in wild-type embryos [3, 16, 32–40], we only identified two studies reporting DA diameters in Endoglin-deficient embryos [3, 37]. Overall, our measurements of DA geometry are in line with the high variability found in the literature (see S6 Table): The majority of the reported means/medians of luminal area [16] and luminal diameter [3, 34, 36, 37, 39, 40] maximally differ by 20% from our measurements. Also, the single observations of wild-type DA diameters reported in [32] are within the range of our measurements. Notably smaller diameters in wild-type DAs at 48 hpf and 72 hpf were reported in [34, 36, 38] and [35], respectively. The authors in [33] measured drastically bigger diameters in wild-types at 72 hpf.

We consider two potential explanations for the observed differences in DA diameter: Firstly, vessel diameters are typically measured in 2D slices or projections of 3D images. In these planar views, it is difficult to ensure that diameter measurements are perpendicular to

the vessel's anterior-posterior (and left-right) axis. But this is necessary as the greatest luminal extension is typically found in dorsal-ventral direction. Secondly, differences between the developmental stages of the analyzed zebrafish embryos can impact geometric measurements: In any study, the process of imaging an entire group of zebrafish embryos takes several hours. These delays during image acquisition increase between-embryo variability. This is especially critical as drastic changes in DA diameter occur during the short time interval analyzed here [3].

Data on EC surface morphology in the DAs of wild-type and Endoglin-deficient embryos are scarce (see S7 Table). This is due to the necessity of mathematical approaches to accurately compute morphometric measures like cell surface area. In [3], unrolling of single EC contours allowed the authors to quantify 2D EC morphology in DAs of wild-type and Endoglin-deficient embryos at 48 hpf and 72 hpf. While the majority of their mean surface areas and perimeters maximally differ by 20% from our mean measurements, they reported (30%–31%) larger mean surface areas at 72 hpf. Their greater surface areas at this time point could potentially be explained by differences between the developmental stages of the analyzed embryos (discussed above), differences in the experimental setup or uncertainty during manual annotation.

Computing unrolled maps of wild-type DAs allowed the authors in [14] to quantify 2D EC morphology at 48 hpf. Their (46%) larger mean EC surface area can partly be explained by the authors' separation of ECs and potentially hemogenic cells based on the cells' morphology and position within the DA; mean area of the latter cells differed less than 20% from our ECs. Additionally, their 2D representation of the DA cannot accurately account for radial and translational variation of vessel geometry. The hereby resulting distortion of 2D contours might have affected computations of surface area.

Although the authors in [16] did not leverage information on EC contours in their approach, they were able to compute averaged EC surface areas from the DA's surface area and the number of EC nuclei within the DA. While many of their mean surface areas in wild-types are in line with ours, they repeatedly measured (more than 20%) smaller mean surface areas at 48 hpf. Our larger mean EC sizes at this time point could be explained by our decision to exclude small round ventral cells during annotation. As these cells are likely to be extruding from the DA, they do not contribute to vessel geometry over a substantial time span.

Finally, the authors in [38] measured drastically (at least 67%) lower mean EC surface areas in wild-type zebrafish at 48 hpf than we did. These large differences can be explained by the authors' approximation of EC surface area based on 2D microscopy slices. Analyzing EC surface area in a planar view neglects the cells' curvature and thus results in smaller area measurements. To limit errors made using this approximation, the authors excluded all ECs that wrapped out of view of the analyzed slice; these (large) cells however strongly impact mean area measurements.

In our analysis we quantified a previously unrecognized dorsal-ventral asymmetry of EC morphology in both wild-type and Endoglin-deficient DAs. Two distinct mechanisms could explain this dorsal-ventral asymmetry: The cell morphology of dorsal and ventral cells could be regulated or constrained by their respective biomechanical environment, e.g., by the presence of the dorsally located stiff notochord or the ventrally located compliant caudal vein (see the mechanical model in [16]). Alternatively, it is possible that dorsal and ventral ECs experience different levels of wall shear stress (WSS). In [41], it was shown that endothelial overexpression of the flow-sensitive transcription factor Klf2a decreased endocardial cell size in zebrafish embryos. Hence, if ventral ECs in the DA experienced higher levels of WSS, this could explain their smaller cell size. This hypothesis is further supported by our finding that, in both phenotypes, ventral ECs were more strongly elongated in the direction of blood flow than dorsal ECs, a process that is mediated by WSS [42]. It is up to follow-up live imaging

studies to investigate whether there exists a correlation or even a causation between the observed dorsal-ventral asymmetry of EC morphology in the DA and local levels of wall shear stress in wild-type and Endoglin-deficient embryos. Functional studies could further explore whether the observed morphological differences between dorsal and ventral ECs in Endoglin-deficient embryos are directly linked to distinct local activity levels of Endoglin and, if that is the case, how the protein's activity is affected by local wall shear stress. Insights generated from such studies can aid in finding out whether variability in EC shape also contributes to the onset of human arteriovenous malformations that occur due to a loss of Endoglin [12].

Lastly, we would like to note that our decision to exclude small round ventral cells from the analysis without staining for HSPC markers could have impacted our quantification of a dorsal-ventral asymmetry of EC morphology at 48 hpf. The cells analyzed in this article could include larger (precursor) HSPCs or the excluded cells could include small endothelial cells. Future studies into the spatial organization of the early developing DA should combine morphometric measurements in dorsal and ventral cells with staining for HSPC markers. As the authors in [32] however detected no EHT events after 60 hpf, our quantification of a dorsal-ventral asymmetry at 72 hpf should not be affected by our exclusion of these small ventral ECs.

In summary, we developed a novel mathematical approach that allows reconstruction of the zebrafish DA's apico-luminal surface using only information from EC contours. By consistently describing vessel geometry and EC morphology, our approach can generate new insights into spatial variability in EC morphology and the relationship between EC morphology and vessel geometry. We demonstrated this ability by applying our approach to wild-type and Endoglin-deficient zebrafish embryos where we identified a previously unrecognized dorsal-ventral asymmetry of EC morphology. Notably, we found that dorsal ECs contributed most to the vessel diameter increase that occurs in Endoglin-deficient embryos between 48 hpf and 72 hpf.

## Supporting information

**S1 Appendix. Supporting methods.** The text references S2–S6 Figs.
(PDF)

**S1 Tutorial. Inspecting endothelial cell surface meshes using Paraview.**
(PDF)

**S1 Fig. Dorsal flattening of dorsal aorta.** Examples for dorsal flattening in the DA. All images are transverse views (view towards posterior). Left column: endothelial cell junctions marked by Pecam1-EGFP (cyan). Middle column: Dextran-perfused lumen (red). Right column: merged signals. Note that in the shown cross-sections, the dorsal segment (white arrow) is flattened while the ventral segment is round. Asterisk: Intersegmental vessel. Scale bar: 10 μm. Snapshots in the upper/lower row stem from wild-type embryo 14/15 (validation data) at 72 hpf, respectively.
(EPS)

**S2 Fig. Insertion of points from neighboring cell contours during enrichment.** Point $q \in$ $EC_{\text{anno},i'}$ can lie in multiple cylinders around edges of the current version of the enriched cell contour $EC_{\text{enri},i}$ of cell $i$. To then decide where to insert $q$ into $EC_{\text{enri},i}$, two scenarios are considered: **(A–B)** If $q$ lies in any cylinders without length extension ($\Delta h = 0$), the edge ($p^*$, $(p^*)^+$) of $EC_{\text{enri},i}$ is determined that has the minimal orthogonal distance to $q$. Note the orthogonal projections $\Pi_p^{\text{line}}(q)$ and $\Pi_{p^*}^{\text{line}}(q)$ of $q$ onto the edges ($p$, $p^+$) and ($p^*$, $(p^*)^+$), respectively. Point $q$ is then inserted between the points $p^*$ and $(p^*)^+$. **(C–D)** Alternatively, if $q$ is located near a sharp bend of the contour and is thus only located within cylinder extensions ($\Delta h > 0$) around

edges of $EC_{enri,i}$, the nearest neighbor $p^* \in EC_{enri,i}$ of $q$ is determined. In this example, point $q$ then becomes the new predecessor of $p^*$ since $q$ is closer to $(p^*)^-$ than to $(p^*)^+$. If $(p^*)^+$ were closer to $q$, then $q$ would become the new successor of $p^*$ instead.
(EPS)

**S3 Fig. Coordinate system and mean shape.** The basis vector $b_z$ in anterior-to-posterior direction defines the set of orthogonal cross-sectional planes with basis vectors $b_x$ in left-to-right direction and $b_y$ in ventral-to-dorsal direction. All (blue) points on EC contour splines were projected onto the $xy$-plane to estimate the mean shape. The (purple) midpoint of the mean shape defines the $xy$-coordinates of the origin of the coordinate system.
(EPS)

**S4 Fig. Projection of cell contour spline onto vessel surface.** The projection of a segment of the cell contour spline $s_{transf,i}$ (gray) onto the estimated vessel surface is illustrated. **(A)** For the pair of points $p$ and $p^+$ on the spline $s_{transf,i}$, the point $s_{transf,i}(u^*)$ on the spline between $p$ and $p^+$ is determined that lies on the cross-sectional plane at $z_k$. The same is done for the cross-sectional plane at $z_{k+1}$, as it also lies between $[p]_z$ and $[p^+]_z$. **(B)** The points $s_{transf,i}(u^*)$ are then projected onto the closest pre-computed points $q^* \in CS_k$ or $q^* \in CS_{k+1}$ on the cross-sectional shapes at $z_k$ and at $z_{k+1}$, respectively.
(EPS)

**S5 Fig. Triangulation respecting cell contour edges. (A)** To triangulate the cell surface between cross-sections $CS_k$ at $z_k$ and $CS_{k+1}$ at $z_{k+1}$, first all points of the (red) cell boundary at these cross-sections are collected (here, $p_1, p_2 \in CS_k$ and $q_1, q_2 \in CS_{k+1}$). On each cross-section, a path within the cell surface is constructed that connects the two points at the cell boundary: $(v_1 = p_1, v_2, v_3 = p_3)$ on $CS_k$ and $(w_1 = q_1, w_2, \ldots, w_8 = q_2)$ on $CS_{k+1}$. **(B)** A triangulation is created covering the surface spanned by these points. These triangles always contain exactly two neighboring points on either cross-section and one point on the other cross-section. Here, the red and orange triangles are constructed using Eq (58) and Eq (59) of S1 Appendix, respectively. **(C)** The triangulation is refined by connecting more points with their closest neighbor on the other cross-section. Here, the triangulation is altered such that $v_3 \in CS_k$ is connected to its closest neighbor $\Pi_{k+1}(v_3) = w_5 \in CS_{k+1}$. Note that cell contours can be annotated in clockwise or counter-clockwise orientation with respect to their nucleus. Thus, from (directed) cell contour edges alone, it is not possible to distinguish whether the cell surface is located on the left or on the right side of these edges. Hence, the complementary vessel surface, which contains the light-gray points, is also triangulated. The cell surface is then extracted from the full vessel surface triangulation in a second step (see S6 Fig).
(EPS)

**S6 Fig. Extraction of triangulation within cell contour.** Beginning with the starting triangle $\Delta$, all neighboring triangles $\Delta' \in N_\Delta$ are added to the cell surface mesh, provided they do not lie outside the cell contour. This process is then repeated: Once a triangle is added to the cell surface mesh, for all its neighbors it is checked whether they lie within the cell contour. Note the two triangles at the top with a common edge touching the cell contour. One of these triangles is added to the cell surface mesh (accepted), the other one is rejected.
(EPS)

**S7 Fig. Manual annotation of 3D confocal microscopy images in Imaris. (A)** Top: Manual annotation of an endothelial cell contour. Points (yellow) were manually placed in the center of the Pecam1-EGFP-marked cell junction signal (cyan) using the "Measurements" tool. **(B)** Manual annotation of coordinate system axes. Approximate directions of anterior-posterior

axis, dorsal-ventral axis and left-right axis were annotated by two points each using the "Measurements" tool. **(C)** Manual annotation of angiogram slices. With the help of the "Ortho slicer" tool, points (magenta) were placed on the intersection of the signal from the Dextran-perfused lumen (red) and the Pecam1-EGFP-marked cell junction signal (cyan) on cross-sectional slices using the "Measurements" tool. Transverse view (dorsal at the top; view towards anterior). The 3D microscopy images in (A) and (B) stem from Endoglin-deficient embryo 13 at 72 hpf (analysis data); the images in (C) stem from wild-type embryo 14 at 72 hpf (validation data).
(EPS)

**S8 Fig. Intermediate results of endothelial cell contour processing.** Using our novel mathematical approach we processed endothelial cell contours as follows: First, endothelial cell contours were manually outlined on Pecam1-EGFP-marked cell junctions **(A)**. These contours were then enriched by points from neighboring cells and by linear interpolation **(B)**. The result of enrichment by neighbors is visible from the additional points on the segment (1) of the blue cell in (B) in comparison to (A); these points originate from the purple cell. The effect of linear interpolation can be seen by the increased number of points on the segment (2) of the blue cell in (B) in comparison to (A). Next, periodic smoothing splines were fitted to the enriched contours and equidistant points were computed on these curves **(C)**. This improved the contours' smoothness, as seen when comparing the segment (3) of the brown cell in (B) and (C). After vessel surface reconstruction, the cell contour splines were finally projected onto the estimated vessel cross-sections **(D)**. Contour projection ensured consistency between estimated vessel cross-sections and endothelial cell contour splines. Points protruding too far into the lumen were moved outwards; this can be seen when comparing the shared segment (5) of the red and grey cells in (C) and (D). Points located too far away from the lumen were moved inwards. For this, compare the segment (4) of the cyan cell in (C) and (D). Note that all contours are shown in the same coordinate system that was estimated from the endothelial cell contour splines. Left images: lateral view (anterior to the left); right images: transverse view (dorsal at the top). The shown cell contours stem from Endoglin-deficient embryo 13 at 72 hpf.
(EPS)

**S9 Fig. Coordinate system within mounting medium vs. estimated coordinate system.** Shown are the endothelial cell contour splines in the coordinate system given by the embryo's position within the mounting medium, i.e., the coordinate system provided by Imaris **(A)**, and in the coordinate system estimated from the endothelial cell contour splines **(B)**. In a lateral view of the DA (left images; anterior to the left), it appears that the direction of the DA's anterior-posterior axis is comparably well-aligned with the $x$-/$z$-axis in the original and estimated coordinate system, respectively. However, differences between the two coordinate system become apparent in a cross-sectional view: Within the estimated coordinate system, the vessel's left-right and dorsal-ventral axes are properly aligned with the $x$- and $y$-axes (right image in **(B)**). In contrast, in the original coordinate system the $z$- and $y$-axes are not properly aligned with the vessel's left-right and dorsal-ventral axes (right image in **(A)**). The shown cell contours stem from Endoglin-deficient embryo 13 at 72 hpf.
(EPS)

**S10 Fig. Iterative choice of tuning parameter values.** We chose tuning parameter values in the following order: 1) the minimal number $n_{oct}$ of points with non-zero weight per plane octant, 2) the upper bound $\lambda$ on the allowed deviation of the locally estimated cross-sectional shape from the mean shape and 3) the standard deviation $\sigma$ of the underlying Gaussian function employed during smoothing of cross-sectional shapes along the vessel axis. **(A–C)**

Goodness of fit: distances between the manually annotated cell contours $EC_{anno}$ and their projections $EC_{proj}$ onto the estimated vessel surfaces for different combinations of values of $n_{oct}$, $\lambda$ and $\sigma$ (see Eq (36) within main text). In the shown distributions, all ECs of wild-type and Endoglin-deficient embryos at 48 hpf and 72 hpf were pooled. Lower bound on contour distances: $n_{oct} = 1$ (orange box plot in (A)); upper bound on contour distances: $\lambda = 0\%$ (mean shape; red box plot in (B)). The numbers below each box plot denote the corresponding numbers of cells. **(D)** Physiological plausibility: exemplary local cross-sectional fit in embryo 9 at 72 hpf. Solid and dashed blue curve: estimated and smoothed local cross-sectional shape for chosen tuning parameters, respectively. Solid and dashed cyan curve: estimated and smoothed local cross-sectional shape for unconstrained estimation, respectively. For additional information, see Fig 4 (main text). **(E)** Smoothness: lateral view (anterior to the left) of estimated vessel surface in embryo 9 at 72 hpf for different sets of tuning parameter values. Blue/cyan lines mark the region of the vessel surface containing the cross-sectional shape shown in (D). $\lambda =$ NA: constraint from Eq (18) (main text) omitted. $\sigma =$ NA: vessel surface not smoothed. Our choice of $n_{oct} = 30$ resulted in small contour distances in comparison with the lower and upper bounds: median contour distance using $n_{oct} = 30$ was only 15% higher than the median contour distance obtained using $n_{oct} = 1$ (compare blue and orange box plots in (A)), but 69% *lower* than the median contour distance obtained using the mean shape along the entire vessel axis (compare blue box plot in (A) with red box plot in (B)). The chosen value for $n_{oct}$ was the largest value that was applicable to all data sets, i.e., embryos at either time point. In the data set that limited the maximal choice of $n_{oct}$, relatively poor quality of the fluorescence signal allowed us only to annotate a single cell on the left side with confidence. As the quality of the fluorescence signal of that data set was good on the right side, we opted not to exclude it. The lower the value of $n_{oct}$, the more frequently vessel cross-sections with a non-physiological shape were estimated due to an insufficient number of points in some sectors of the cross-sectional plane. For example, compare the cross-sectional irregularities in the most anterior cross-sections (marked by purple line) in panel (i) of (E) to the same cross-sections in panel (ii). Although constraining the estimation of cross-sectional shapes using $\lambda = 20\%$ slightly increased the median contour distance by 4.0% (compare blue box plots in (A) and (B)), it visibly improved the physiological plausibility of cross-sectional shapes in regions of the anterior-posterior axis where large angular sectors of the cross-sectional plane contained only limited information on cell contours. An example of this can be seen in the dorsal segments of the solid blue and solid cyan curve in (D) where the estimation of the local cross-sectional shape was only informed by far-away points (with low weights). Although smoothing of cross-sectional shapes with $\sigma = 10$ μm further increased the median contour distance by 70% (compare blue box plots in (B) and (C)), the final median contour distance was still 45% lower than the median contour distance obtained using the mean shape along the entire vessel axis (compare blue box plot in (C) with red box plot in (B)). Importantly, this value visibly improved the smoothness of the estimated vessel surfaces (compare panels (iii) and (iv) of (E)). Note that the combination of constrained estimation and smoothing was necessary to obtain an accurate reconstruction of the DA of embryo 9 at 72 hpf. For example, compare dorsal segments of dashed blue and dashed cyan curve in (D) and compare region marked by red line in panels (iv) and (v) of (E).
(EPS)

**S11 Fig. Compliance of estimated cross-sectional shapes with angiography.** Distributions of the relative cross-sectional deviations $dev_{cross}$ of estimated vessel surfaces from angiogram slices for two wild-type embryos at 72 hpf with two sets of cell contour annotations each (validation data). Vessel surface slices estimated from either of the two versions of annotated cell

contours pooled per embryo. Deviations computed with Eq (16) (main text). Distributions stratified by **(A)** the cross-sectional shape used during estimation or, **(B)** when employing the superellipse shape, the number of annotated cells located within a distance of 0.5 μm to the slicing plane. The numbers below each box plot denote the corresponding numbers of vessel surface slices. Using either circular, elliptic or superelliptic cross-sections, the median deviation of vessel surface slices from the angiogram slices was lower than 19% in each of the two embryos (A). Overall, ellipses and superellipses described angiogram slices comparably well; both shapes were more accurate than circles: In embryo 14, circular cross-sections had 14%/22% higher median deviations from the angiogram slices than elliptic/superelliptic cross-sections, respectively. In embryo 15, median deviations of circular cross-sections were 23%/13% higher than those of elliptic/superelliptic cross-sections, respectively. When using the superellipse shape, cross-sections estimated in positions where only one cell was annotated, were not guaranteed to closely resemble the angiogram slices (B): In embryo 14, the median deviation of the estimated cross-sections from the angiogram slices was only 8.4% higher for angiogram slices with only one nearby cell in comparison to slices with more cells. In embryo 15, however, the median cross-sectional deviation was 75% higher for angiogram slices with only one nearby cell.
(EPS)

**S12 Fig. Correlations between morphometric measures.** Pair-wise correlations between morphometric measures of ECs. Spearman's rank correlation coefficient $r_s$ computed per panel. Analysis based on ECs with projection distances (onto the estimated vessel surfaces) comparable to or lower than the maximal annotation uncertainty of 0.742 μm. Cell surface area and perimeter showed a strong positive correlation. Also, when separating cells by phenotype and time, this correlation remained strong ($0.784 \leq r_s \leq 0.916$). This meant that ECs had few protrusions, otherwise large increases of perimeter could have been accompanied by small increases in surface area. Elongation and compactness had a moderately strong negative correlation. When separating cells by phenotype and time, this correlation was slightly weaker ($-0.562 \leq r_s \leq -0.466$). In the absence of many protrusions, this negative correlation implied that very elongated cells (with low compactness) had a tendency to be aligned in the direction of flow, i.e., they had a high value of our elongation measure.
(EPS)

**S13 Fig. Within-embryo variability in endothelial cell morphology larger than within-embryo variability in luminal area.** Pair-wise correlations between within-embryo variability, i.e., quartile coefficient of dispersion (QCD), of luminal area and QCD of morphometric measurements. Each point corresponds to a data set, i.e., an embryo at either time point. Analysis based on cross-sections with at least two annotated cells present within a distance of 0.5 μm to the cross-sectional plane and ECs with projection distances (onto the estimated vessel surfaces) comparable to or lower than the maximal annotation uncertainty of 0.742 μm. In the vast majority of data sets, QCD of morphometric measurements was more than 50% larger than QCD of luminal area in the same embryo at the same time point: this occurred in 25 of 26 data sets for EC surface area, in 19 of 26 data sets for EC perimeter, in 23 of 26 data sets for EC compactness and in 26 of 26 data sets for EC elongation.
(EPS)

**S14 Fig. Within-embryo variability in cell surface area and elongation larger than within-embryo variability in perimeter and compactness.** Pair-wise correlations between within-embryo variability, i.e., quartile coefficient of dispersion (QCD), of morphometric measures. Each point corresponds to a data set, i.e., an embryo at either time point. Analysis based on

ECs with projection distances (onto the estimated vessel surfaces) comparable to or lower than the maximal annotation uncertainty of 0.742 μm. Within-embryo variability was largest in EC surface area and EC elongation: In 22 of 26 and 17 of 26 data sets, QCD of EC surface area was more than 25% larger than QCD of perimeter and compactness, respectively. In 20 of 26 data sets each, QCD of EC elongation was more than 25% larger than QCD of perimeter and compactness.
(EPS)

**S15 Fig. Morphological characteristics of non-classified endothelial cells.** Extended version of Fig 11 (main text). Distributions of morphometric measurements over all ECs, stratified by location and time, for wild-type embryos (left column) and for Endoglin-deficient embryos (right column). Analysis based on ECs with projection distances (onto the estimated vessel surfaces) comparable to or lower than the maximal annotation uncertainty of 0.742 μm. The numbers below the bottom box plots denote the corresponding numbers of cells for each column. In each phenotype and at each time point, non-classifiable ECs (denoted by NA) had the highest median cell size **(A–D)**, their median compactness hardly differed from the classified cells **(E–F)** and they had the lowest median elongation in the direction of flow **(G–H)**.
(EPS)

**S16 Fig. Dorsal-ventral asymmetry of endothelial cell morphology based on more detailed classification.** Distribution of morphometric measurements over all ECs, stratified by detailed location: Cells can be located almost exclusively in a single cross-sectional sector, i.e., have > 75% of their surface area in the dorsal sector (D), either the left or right sector (LR), or in the ventral sector (V). Cells wrapping around a larger fraction of the vessel, e.g., D-LR and LR-V, have an area contribution of > 50% but ≤ 75% in the first sector; the majority of their remaining surface area is located in the second sector(s). Left column: wild-type embryos; right column: Endoglin-deficient embryos. Analysis based on ECs with projection distances (onto the estimated vessel surfaces) comparable to or lower than the maximal annotation uncertainty of 0.742 μm. The numbers below the bottom box plots denote the corresponding numbers of cells for each column; box plots for groups with fewer than 4 observations not drawn. Among the differently classified cells, two groups stuck out: (1) the dorsal subpopulation D-LR with substantial area contribution in the left/right sectors and (2) the almost exclusively ventral subpopulation V. Comparing these two groups of cells, we found that D-LRs had larger surface areas in both phenotypes at each time point **(A–B)**: In wild-types, median surface area of D-LRs was 50%/20% higher than that of Vs at 48 hpf/72 hpf, respectively. In Endoglin-deficient embryos, these differences were even more pronounced. Here, D-LRs had a 66%/96% higher median area than Vs at 48 hpf/72 hpf, respectively. We found that changes in surface area over time were more pronounced in D-LRs; this was most noticeable in Endoglin-deficient embryos: In wild-types, median cell surface area of D-LRs decreased by 19% between 48 hpf and 72 hpf, while it *increased* by 66% in Endoglin-deficient embryos. Median cell surface area of Vs barely increased by 1.9% in wild-types, but *noticeably* increased (by 41%) in Endoglin-deficient embryos. Perimeters were mostly comparable between the two groups **(C–D)**: In wild-types at each time point and in Endoglin-deficient embryos at 48 hpf, median perimeter of D-LRs maximally differed by 15% from that of Vs. In Endoglin-deficient embryos at 72 hpf however, D-LRs had a 31% higher median perimeter than Vs. We found that D-LRs had higher compactness than Vs **(E–F)** but the magnitude of this difference strongly varied: In wild-types at 48 hpf, median compactness of D-LRs was 2.1%/31% higher than that of Vs at 48 hpf/72 hpf, respectively. In Endoglin-deficient embryos, D-LRs had a 16%/6.8% higher median compactness at 48 hpf/72 hpf, respectively. Finally, D-LRs were less elongated in the direction of flow than Vs **(G–H)**: In wild-types at 48 hpf, median elongation of D-LRs was 43%/42%

lower than that of Vs at 48 hpf/72 hpf, respectively. In Endoglin-deficient embryos, D-LRs had a 42%/51% lower median elongation at 48 hpf/72 hpf, respectively. We found that changes in elongation over time were less pronounced in D-LRs; this was most noticeable in Endoglin-deficient embryos: In wild-types, median elongation of D-LRs increased by 100% between 48 hpf and 72 hpf, while it barely changed (by 9.3%) in Endoglin-deficient embryos. Median elongation of Vs increased by 99.2% in wild-types but only by 29% in Endoglin-deficient embryos. (EPS)

**S17 Fig. 2D morphology of dorsal and ventral endothelial cells.** Vessel surfaces (analysis data) were reconstructed using the mean shape only, i.e., setting $\lambda = 0\%$. For each endothelial cell, it was then attempted to classify it as either dorsal, ventral or left/right (using the simple classification from Section 9.4 of S1 Appendix) and its morphology was quantified. Next, these cells were unrolled into 2D representations (see Section 11 of S1 Appendix). Shown here are all unrolled surfaces of cells classified as either dorsal or ventral. In contrast to the analysis in the main text, cells with high contour projection distances not excluded. Cells grouped by their classification, developmental stage (48 hpf or 72 hpf) and by phenotype (wt or Eng-def). Cells in each group sorted by cell surface area. The horizontal component of each panel is the DA's anterior-posterior axis, i.e., the direction of blood flow; the vertical component is the circumference of the DA. ⁻/⁻⁻/⁻⁻⁻: The cell's projection distance when using the mean shape is greater than 1/2/3 times its projection distance when employing the main text's tuning parameter values. When inspecting ECs in wild-type embryos, we made the following observations: At 48 hpf, surface areas were higher in dorsal ECs (median: 909 $\mu m^2$) than in ventral ECs (median: 663 $\mu m^2$). At 72 hpf however, dorsal ECs had similar surface areas (median: 602 $\mu m^2$) as ventral ECs (median: 633 $\mu m^2$). Note that when using different tuning parameter values and excluding cells with high projection distances, we found that dorsal ECs were *larger* than ventral ECs at 72 hpf (see Fig 11A within main text). At both time points, dorsal ECs were less elongated in flow direction (medians: 1.20/2.60 at 48 hpf/72 hpf) than ventral ECs (medians: 2.12/4.09 at 48 hpf/72 hpf). When inspecting ECs in Endoglin-deficient embryos, we obtained the following results: At 48 hpf, surface areas were similar in dorsal ECs (median: 585 $\mu m^2$) and ventral ECs (median: 580 $\mu m^2$). At 72 hpf however, surface areas of dorsal ECs (median: 1150 $\mu m^2$) were much higher than those of ventral ECs (median: 686 $\mu m^2$). Between 48 hpf and 72 hpf, elongation in flow direction hardly changed in dorsal ECs (medians at 48 hpf/72 hpf: 1.39/1.45) but increased in ventral ECs (medians at 48 hpf/72 hpf: 2.09/2.92). Note that in total 7 out of 296 cells were classified differently when using the mean shape and not the tuning parameter values employed in the main text. Also, the increases in contour projection distance when using the mean shape were substantial for some of the analyzed cells (see cells labeled with ⁻⁻/⁻⁻⁻). However, their morphometric measurements were still comparable (compare with rows where $\lambda = 0\%$ in S4 Table). (EPS)

**S1 Table. Frequently used mathematical symbols.**
(PDF)

**S2 Table. Identified tuning parameter values.**
(PDF)

**S3 Table. Robustness of chosen tuning parameter values w.r.t. cross-sectional geometry.** Reference vessel cross-sectional shapes estimated with $n_{\mathrm{oct}} = 30$, $\lambda = 20\%$ and $\sigma = 10$ $\mu m$. Each vessel cross-sectional shape re-estimated after altering either $n_{\mathrm{oct}}$, $\lambda$ or $\sigma$, while fixing the other two parameters. Relative deviations $\mathrm{dev}_{\mathrm{cross}}$ of the re-estimated vessel cross-sectional shapes from the reference shapes computed with Eq (16) (main text); results for pooled vessel cross-

sectional shapes from wild-type and Endoglin-deficient embryos at 48 hpf and 72 hpf. When $n_{oct} = 60$, the number of cross-sections was 24 265. In all other computations, the number of cross-sections was 25 111. $^{\dagger}$: constraint from Eq (18) (main text) omitted. $^{*}$: vessel surface not smoothed. When halving or doubling either $n_{oct}$, $\lambda$ or $\sigma$, the majority of estimated vessel cross-sections had a relative deviation of less than 5%. Notably, the largest median differences were found when using the mean shape only ($\lambda = 0\%$), followed by the setting where smoothing was omitted. The largest maximal differences were obtained when not constraining the deviation of the local cross-sectional shape from the mean shape.
(PDF)

**S4 Table. Robustness of chosen tuning parameter values w.r.t. endothelial cell morphology.** Reference cell morphometric measurements computed after vessel surface reconstruction with $n_{oct} = 30$, $\lambda = 20\%$ and $\sigma = 10$ μm. Morphology of each cell re-computed after altering either $n_{oct}$, $\lambda$ or $\sigma$, while fixing the other two parameters. Relative deviations $dev_{rel}$ of re-computed cell morphometric measurements from their reference values computed with Eq (37) (main text); results for pooled EC surfaces from wild-type and Endoglin-deficient embryos at 48 hpf and 72 hpf. $f$: relative frequency. Note: We omitted the column $f(dev_{rel} < -10\%)$ as it can be computed as $f(dev_{rel} < -10\%) = 100\% - f(|dev_{rel}| \leq 10\%) - f(dev_{rel} > 10\%)$. When $n_{oct} = 60$, the number of cells was 290. In all other computations, the number of cells was 296. $^{\dagger}$: constraint from Eq (18) (main text) omitted. $^{*}$: vessel surface not smoothed. When halving or doubling either $n_{oct}$, $\lambda$ or $\sigma$, each morphometric measurement differed by maximally 10%. Notably, when not smoothing, larger cell surface areas were computed. As a consequence, elongation was decreased and compactness increased.
(PDF)

**S5 Table. Compliance of luminal areas of estimated cross-sectional shapes with angiography.** Relative deviations $dev_{rel}$ of the luminal areas of slices of the estimated vessel surfaces from angiogram slices for two wild-type embryos at 72 hpf with two sets of cell contour annotations each (validation data), computed with Eq (37) (main text). Vessel cross-sections estimated from either of the two versions of annotated cell contours pooled per embryo. $f$: relative frequency. Note: We omitted the column $f(dev_{rel} < -10\%)$ as it can be computed as $f(dev_{rel} < -10\%) = 100\% - f(|dev_{rel}| \leq 10\%) - f(dev_{rel} > 10\%)$. The number of vessel surface slices was 90 for embryo 14 and 102 for embryo 15. The superellipse performed best for embryo 14 and the ellipse performed best for embryo 15. In embryo 14, more luminal areas were under- than overestimated, in embryo 15 vice versa.
(PDF)

**S6 Table. Comparison to literature-reported geometric measurements of dorsal aorta.** Analysis based on cross-sections with at least two annotated cells present within a distance of 0.5 μm to the cross-sectional plane. The number of vessel cross-sections was 5575/5972 for wild-type embryos and 4682/5185 for Endoglin-deficient embryos at 48 hpf/72 hpf, respectively. $^{*}$: individual measurements weighted by numbers of cross-sectional shapes per embryo (see Section 12 of S1 Appendix). Literature values digitized from graphics using WebPlotDigitizer; except for [33]: values found in text. [38]: Mean and s.d. in S2(J) Fig obscured by data points and could not be digitized. $^{¤}$: absolute relative deviation of literature mean/median value from our measured mean/median value is greater than 20%. s.d.: standard deviation. s.e.m.: standard error of mean. $Q_1$: 25% percentile. $Q_2$: median. $Q_3$: 75% percentile.
(PDF)

**S7 Table. Comparison to literature-reported morphological measurements of endothelial cells.** Analysis based on ECs with projection distances (onto the estimated vessel surfaces)

comparable to or lower than the maximal annotation uncertainty of 0.742 μm. The number of cells was 73/62 for wild-type embryos and 71/68 for Endoglin-deficient embryos at 48 hpf/72 hpf, respectively. *: individual measurements weighted by numbers of cells per embryo (see Section 12 of S1 Appendix). Literature values digitized from graphics using WebPlotDigitizer. [3]: compactness not computed; definition of elongation differed from ours. ¤: absolute relative deviation of literature mean value from our measured mean value is greater than 20%. s.d.: standard deviation. s.e.m.: standard error of mean. EHT: ECs undergoing endothelial-to-hematopoietic transition. EHT?: potentially hemogenic cells.
(PDF)

## Acknowledgments

We would like to thank Anne Kühnel, Angela Hubig, Bärbel Wuntke and Julius Schwarz (University of Potsdam) for support during experiments and data acquisition. Furthermore, we would like to thank Myfanwy Evans (University of Potsdam) and Konrad Polthier and his working group (Freie Universität Berlin) for fruitful and stimulating discussions.

## Author Contributions

**Conceptualization:** Daniel Seeler, Nastasja Grdseloff, Claudia Jasmin Rödel, Salim Abdelilah-Seyfried, Wilhelm Huisinga.

**Data curation:** Daniel Seeler, Nastasja Grdseloff, Claudia Jasmin Rödel.

**Formal analysis:** Daniel Seeler.

**Investigation:** Daniel Seeler, Nastasja Grdseloff, Claudia Jasmin Rödel.

**Methodology:** Daniel Seeler, Wilhelm Huisinga.

**Project administration:** Salim Abdelilah-Seyfried, Wilhelm Huisinga.

**Resources:** Salim Abdelilah-Seyfried, Wilhelm Huisinga.

**Software:** Daniel Seeler.

**Supervision:** Charlotte Kloft, Salim Abdelilah-Seyfried, Wilhelm Huisinga.

**Validation:** Daniel Seeler, Nastasja Grdseloff, Claudia Jasmin Rödel.

**Visualization:** Daniel Seeler.

**Writing – original draft:** Daniel Seeler.

**Writing – review & editing:** Daniel Seeler, Nastasja Grdseloff, Claudia Jasmin Rödel, Charlotte Kloft, Wilhelm Huisinga.

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
