## [Decision Letter · Decision Letter 0]

9 Apr 2024

Dear Prof. Dr. Huisinga,

Thank you very much for submitting your manuscript "Novel mathematical morphology model identifies dorsal-ventral asymmetry of endothelial cell morphology in dorsal aorta of wild-type and Endoglin-deficient zebrafish embryos" for consideration at PLOS Computational Biology.

As with all papers reviewed by the journal, your manuscript was reviewed by members of the editorial board and by several independent reviewers. In light of the reviews (below this email), we would like to invite the resubmission of a significantly-revised version that takes into account the reviewers' comments.

We cannot make any decision about publication until we have seen the revised manuscript and your response to the reviewers' comments. Your revised manuscript is also likely to be sent to reviewers for further evaluation.

Sincerely,

Alison Marsden

Academic Editor

PLOS Computational Biology

Jason Haugh

Section Editor

PLOS Computational Biology

Reviewer's Responses to Questions

**Comments to the Authors:**

Reviewer #1: The work by Seeler et al. uses a newly developed mathematical morphology model to describe endothelial cell shapes within the dorsal aorta of wildtype and endoglin mutant zebrafish embryos. Their model accounts for the dorsal flattening of the dorsal aorta, while previous models assumed the vessel to be a cylinder. The authors also provide a nice visualization tool using paraview. Overall, the data seem to be of great quality and will better enable the analysis of endothelial cell shapes in the future. This said, I was not able to judge or check the mathematics behind their calculations, due to my lack of expertise. However, regarding the biology of the system, I have several specific comments.

1. The authors claim that they have “identified a previously unrecognized dorsal-ventral asymmetry of EC morphology”, as also the title states. However, the authors leave it undiscussed to what extent the smaller ventral cells they observe are hemogenic endothelial cells that are well described in zebrafish (e.g. Kissa and Herbomel, Blood and Nature papers). It is also established that they have a different morphology and are smaller (Lancino et al., elife 2018). The authors state that they excluded “protruding” cells from the DA (hematopoietic stem cells), but they leave it unclear, whether the smaller cells are on their way to become hematopoietic stem cells. Clearly some work on the biology would be necessary here. Do the smaller cells express runx1 or c-myb? Can the authors use transgenic zebrafish to monitor e.g. runx1 expression and then determine cell sizes to see whether smaller cells express runx1?

2. How about statistical analysis? Are the size and other differences the authors measure statistically significant? Can the authors add this to the figures and legends?

3. Are there different EC cell populations (e.g. based on size) in the ventral DA? An examination of the data the authors provide shows that often larger cells enwrap one side of the DA and greatly extend into the ventral area of the DA, while cells that are only located ventrally indeed appear smaller. How do the authors account for this morphology in their analysis?

Reviewer #2: The paper presents a novel mathematical morphology model to study dorsal-ventral asymmetry in the morphology of endothelial cells within the dorsal aorta of zebrafish embryos, both wild-type and those deficient in Endoglin. Endothelial cells, lining blood vessel lumens, undergo shape changes in response to altered blood flow which affecting vessel geometry by elongating in the flow direction and altering the vessel’s diameter. These changes are critical during early embryonic development, particularly between 48 and 72 hours post-fertilization, and require Endoglin—a protein whose absence leads to excessive endothelial cell growth and increased vessel diameter.

This manuscript is generally well describe the mathematical model analyzing dorsal-ventral asymmetry of endothelial cell morphology in the dorsal aorta of zebrafish embryos introduces a novel and sophisticated approach to understanding complex biological structures. Here are some points that could be considered to improve the manuscript. The model's robustness against variations in parameter values should be a standard and clearly defined. For example, why and where the parameters are from. Also, while the paper might present a novel approach, the real test of any model validity is important to accurately describe experimental observations. Questions arises regarding the comprehensiveness of the data used for validation and whether the model can generalize well.

Also, is there any way that the mathematical model can show how endoglin expression affects the morphology of cells or vessel geometry? such as chemical reactions.

Reviewer #3: The authors present a new methodological pipeline that allows for three-dimensional reconstructions of vessel lumens based off the contours of individual endothelial cells. Their method focuses on accurately characterizing the morphology of each individual endothelial cell within the vessel wall environment. The single cell distinction allows the authors to characterize the dynamics of different groups of endothelial cells – ventral vs dorsal vs left/right, as well as consider adjacent lumen vessel cross-sectional area. While the authors methods may fall under a mathematical model in the sense that they capture essential features of a biological system in mathematical terms, the title and framing of their methodological pipeline as a simple ‘mathematical model’ seem misleading as it does not capture the fact that the user of said model must start from microscopy images, contour them and accurately reconstruct them in three-dimensions. Overall, the approach of the manuscript is not hypothesis driven but rather technological, outlining a carefully laid out methodology for measuring key features of dorsal aorta anatomy from the single cell to vessel level that is applied to two organism models (control and mutant zebrafish models) at two different time points. Even though a mutant model is used the paper doesn’t read as if the system was challenged in order to understand underlying biological mechanisms. It reads as a descriptive validation of the methodology. The impact of the paper and its accessibility can be improved by complete re-organization of the figures and more focusing of the text.

Major points:

+ Apart from a small portion of the methodology overview (Fig. 1) there are no in-vivo images of the dorsal aorta or endothelial cells. Even in Figure 1, the individual cells cannot be distinguished. More importantly, it appears that the reader would have to download the whole package suite/all results in order to see a manual outline of an endothelial cell. All cell morphology images are cartoon schematics. Please include at least one annotation image of the cell data within the paper. Would be great to see cell contour enrichment on top of an actual cell for example.

+ The authors frequently refer to ‘good agreement with results reported in [ref]’ without acknowledging why ref was so important (was it the only instance that anyone has measured said value, etc) or what method was used in the ref. (Example page 5 line 87; page 33 line 798, 804, 806, 807).

+ The authors are commended for studying intra-embryo and between embryo variability. Given the complexity and length of the dorsal aorta it seems normal that relative position would play a role in variability, so perhaps it is not necessary to report detailed endothelial cell morphology variations without stratification.

+ so many figures, including the novel findings of dorsal-ventral asymmetry endothelial cell morphology are summarized in box plots (Fig. 11). Please couple these box plots with some images and schematics to really highlight the importance of this methodology and its applications. Page 29 lines 681 -694 are so interesting. It would be great to visualize the phenomenon within. Along these lines it would also be nice to see the ‘dorsal aorta flattening’ observed in 3D microscopy images.

Other comments:

+page 3 lines 19-21 would be nice for the ‘drastic changes’ that the dorsal aorta undergoes from 48hpf to 72hpf to be captured in a schematic

+ For the major challenge of vessel reconstructions with images lying on one side, is it not possible to flip the embryos within their eggs?

**Have the authors made all data and (if applicable) computational code underlying the findings in their manuscript fully available?**

Reviewer #1: Yes

Reviewer #2: Yes

Reviewer #3: Yes

PLOS authors have the option to publish the peer review history of their article (what does this mean?). If published, this will include your full peer review and any attached files.

Reviewer #1: No

Reviewer #2: No

Reviewer #3: No
---

## [Decision Letter · Decision Letter 1]

10 Jul 2024

Dear Prof. Dr. Huisinga,

We are pleased to inform you that your manuscript 'Novel mathematical approach to accurately quantify 3D endothelial cell morphology and vessel geometry based on fluorescently marked endothelial cell contours: Application to the dorsal aorta of wild-type and Endoglin-deficient zebrafish embryos' has been provisionally accepted for publication in PLOS Computational Biology.

Best regards,

Jason M. Haugh

Section Editor

PLOS Computational Biology

Reviewer's Responses to Questions

**Comments to the Authors:**

Reviewer #2: They addressed all my concerns.

**Have the authors made all data and (if applicable) computational code underlying the findings in their manuscript fully available?**

Reviewer #2: None

PLOS authors have the option to publish the peer review history of their article (what does this mean?). If published, this will include your full peer review and any attached files.

Reviewer #2: **Yes: **Juhyun Lee

---

## [Editor Report · Acceptance letter]

25 Jul 2024

PCOMPBIOL-D-24-00293R1 

Novel mathematical approach to accurately quantify 3D endothelial cell morphology and vessel geometry based on fluorescently marked endothelial cell contours: Application to the dorsal aorta of wild-type and Endoglin-deficient zebrafish embryos

Dear Dr Huisinga,

I am pleased to inform you that your manuscript has been formally accepted for publication in PLOS Computational Biology. Your manuscript is now with our production department and you will be notified of the publication date in due course.

With kind regards,

Lilla Horvath
